# SHIELD: a platform for high-throughput screening of barrier-type DNA elements in human cells

Meng Zhang [1,2], Mary Elisabeth Ehmann[1], Srija Matukumalli[3], Aashutosh Girish Boob[1,2], David M. Gilbert [4] & Huimin Zhao [1,2,5] ✉

Chromatin boundary elements contribute to the partitioning of mammalian genomes into topological domains to regulate gene expression. Certain boundary elements are adopted as DNA insulators for safe and stable transgene expression in mammalian cells. These elements, however, are ill-defined and less characterized in the non-coding genome, partially due to the lack of a platform to readily evaluate boundary-associated activities of putative DNA sequences. Here we report SHIELD (Site-specific Heterochromatin Insertion of Elements at Lamina-associated Domains), a platform tailored for the high-throughput screening of barrier-type DNA elements in human cells. SHIELD takes advantage of the high specificity of serine integrase at heterochromatin, and exploits the natural heterochromatin spreading inside lamina-associated domains (LADs) for the discovery of potent barrier elements. We adopt SHIELD to evaluate the barrier activity of 1000 DNA elements in a high-throughput manner and identify 8 candidates with barrier activities comparable to the core region of cHS4 element in human HCT116 cells. We anticipate SHIELD could facilitate the discovery of novel barrier DNA elements from the non-coding genome in human cells.

Mammalian genomes are spatiotemporally organized into functionally distinct domains that are essential for regulating gene expression[1]. The partitioning of genomes into alternating repressive and active domains relies on boundary elements that delineate heterochromatic and euchromatic regions to prevent crosstalk between adjacent domains[2,3]. Importantly, the disruption of boundary regions can lead to gene misexpression and cause disease[4]. Moreover, certain boundary elements have been widely adopted for stable and safe transgene expression in mammalian cells[5–7]. However, despite the biological significance and engineering applications of boundary elements, our understanding of them remains limited compared to other standard genetic elements (e.g., promoters, enhancers) in the non-coding genome[4].

Boundary elements may possess two distinct functions: enhancer-blocking activity to prevent enhancer-dependent gene activation, or barrier activity to block heterochromatin encroachment[5,8,9]. Although numerous computational tools were developed to predict these elements in silico[10–13], only a few predicted candidates underwent experimental validation which focused exclusively on the enhancer-blocking activity. In contrast, the barrier activity is less examined largely due to the lack of a platform to readily evaluate the ability of DNA sequences to block heterochromatin spreading in mammalian cells. As a result, mammalian barrier elements are ill defined and their associated sequence features remain elusive[4].

To overcome this bottleneck, we seek to develop a high-throughput platform for barrier activity screening in human cells.

[1]Department of Chemical and Biomolecular Engineering, University of Illinois at Urbana-Champaign, Urbana, IL 61801, USA. [2]Carl R. Woese Institute for Genomic Biology, University of Illinois at Urbana-Champaign, Urbana, IL 61801, USA. [3]Department of Biochemistry, University of Illinois at Urbana-Champaign, Urbana, IL 61801, USA. [4]San Diego Biomedical Research Institute, San Diego, CA 92121, USA. [5]Department of Chemistry, Department of Bioengineering, University of Illinois at Urbana-Champaign, Urbana, IL 61801, USA. ✉e-mail: zhao5@illinois.edu

Toward this goal, we choose to exploit the natural spreading of heterochromatin inside the lamina-associated domain (LAD), i.e., the "gene-silencing hub" in mammalian genomes[14,15]. We hypothesize that a carefully selected LAD locus with a strong repressive epigenetic landscape could rapidly silence a reporter gene, which, when shielded by potent barrier elements, would retain active expression. Hence, by directly challenging candidate elements at the same highly repressive LAD locus, their barrier activities could be systematically evaluated without being subject to chromosome position effects[9,16]. However, such a strategy requires an efficient means of inserting DNA into the highly compact heterochromatin, which is typically difficult to edit in the genome[17,18]. Moreover, given the limited understanding of silencing kinetics in mammalian cells, the epigenetic features that dictate silencing rate remain unclear, especially considering the heterogeneity of LADs[19]. Therefore, to facilitate the screening of barrier (or the anti-silencing) activity of DNA elements, it is also needed to establish a platform to monitor transgene silencing at epigenetically distinct loci.

Here we present SHIELD (Site-specific Heterochromatin Insertion of Elements at Lamina-associated Domains), a high-throughput platform tailored for the screening of barrier-type DNA elements in human cells. Built upon the high specificity of PhiC31 integrase at compact heterochromatin as we discovered in this work, SHIELD could achieve successful insertion of plasmid-sized DNA fragments into highly repressive LAD loci with high efficiency and fidelity, thus enabling the screening of barrier DNA elements in a high-throughput manner. We reveal three kinetic classes of gene silencing depending on the local epigenetic landscape, and report a LAD-induced silencing pattern distinct from the all-or-none silencing phenomenon. We further adopt SHIELD to evaluate the barrier potential of 1000 DNA elements (250 bp each) in human cells and identify 8 candidates with activities comparable to or better than the core region of the insulator element cHS4 (chicken β-globin hypersensitive site 4). Our results indicate transcription factors USF and VEZF1 are likely important players in establishing chromatin boundary, and highlight the underappreciated role of mammalian-wide interspersed repeats (MIRs) as potent barrier elements in addition to CTCF-based chromatin insulators.

## Results

### Design of the SHIELD platform
To achieve systematic screening of barrier elements, we opted for *targeted* integration as opposed to the classical barrier assay that relies on *random* insertion of reporter DNA[8,9]. For integration sites, we focused on LADs (Fig. 1a) because their repressive nature may induce epigenetic silencing with fast onset, as opposed to the classical barrier assay where silencing emerges relatively slowly over time[8]. To achieve efficient integration of reporters at heterochromatin, we adopted the large serine integrase-based landing pad strategy[20,21] (Supplementary Fig. 1) as discussed below. Following the establishment of chassis cell lines, a donor plasmid carrying the reporter gene (i.e., enhanced green fluorescent protein, *EGFP*) is integrated at the preselected LAD locus (Fig. 1a) by SHIELD. Stable integrants with on-target insertion are selected with puromycin (Supplementary Fig. 1B). Removing puromycin then allows cells with epigenetically silenced reporters to propagate over time. The polyclonal cells are sorted by fluorescence-activated cell sorting (FACS) based on *EGFP* expression levels at certain time points following the removal of selection pressure. If flanked by potent barrier elements, *EGFP* would be shielded from epigenetic silencing, resulting in active expression. Hence, by tracking the enrichment of library DNA elements in the sorted population by next-generation sequencing (NGS), their relative barrier activities can be determined in a high-throughput manner (Fig. 1a).

### Selection of candidate loci with distinct epigenetic features
To identify integration sites susceptible to epigenetic silencing, we started with five heterochromatin loci (region 6-10) previously reported[22]. We excluded region 8 and region 9 due to the reported low editing efficiency of CRISPR/Cas9 at these two sites[22]. The three remaining loci were renamed as follows: H1 (region 6), H2 (region 7) and H3 (region 10). One housekeeping gene E1 (*Hsp70*) was included as a control for euchromatin. Chromosome coordinates, guide RNA sequences and genotyping primers for Cas9-mediated knock-in (KI) of landing pads at four target sites are listed in Supplementary Table 1.

Figure 1b provides an overview of the local epigenetic landscape (~400 kb) at each locus in the human HCT116 cell line. We focused on the LMNB1 DamID signal track that measures the interacting frequency of local chromatin with nuclear lamina, which has been shown as an repressive nuclear compartment in mammalian cells[23,24]. In addition, we included histone marks H3K9me3 and H3K27me3 as they are also indicative of epigenetic repression[25,26]. Based on these marks, H1 appears as the most repressive site as it resides inside a constitutive LAD with high LMNB1 DamID signal and the most H3K9me3 marks nearby (i.e., constitutive heterochromatin), whereas H2 and H3 are less repressive in this regard. Interestingly, H3 is located not inside a LAD but at its boundary (Fig. 1b, H3), and is likely part of the facultative heterochromatin considering the enriched H3K27me3 histone mark[26]. In fact, H3 is upstream of the human β-globin gene (*HBB*) that is actively expressed in erythroid cells but silenced in other cell types including colorectal carcinoma cells HCT116 (Fig. 1b, H3: RefSeq and RNA-Seq), in line with the dynamic feature of facultative heterochromatin. In addition to histone marks, we also included the DNaseI HS (DNase I hypersensitive site) information to reflect the local chromatin compactness. We noticed more DNaseI HSs within the ~400 kb window shown for H2/H3 than H1, indicating a less condensed chromatin structure at H2/H3 than at H1.

By contrast, E1 exhibits typical euchromatic features (Fig. 1b), including high transcription activity at a gene-dense region, enriched active histone marks (e.g., H3K4me1 and H3K27ac), highly prevalent DNaseI HSs, and the lack of repressive histone marks (e.g., H3K9me3 and H3K27me3). Based on these analyses, we provide a simplified schematic showing the relative levels of epigenetic repression and transcription activity at four sites (Fig. 1c).

We also examined the endogenous DNA methylation at these regions by including the published methylation-reduced representation bisulfite sequencing (methyl-RRBS track) data. We found abundant CpG islands around E1 but much fewer at three heterochromatin sites (Fig. 1b). This difference in CpG density is likely due to the difference in gene density between euchromatin and heterochromatin, as CpG islands are typically associated with promoters of genes. Specifically, the majority of CpG dinucleotides near E1 are not methylated (green), indicating an active transcription environment. Although the remaining CpG dinucleotides at E1 appear moderately (orange) or highly methylated (red), most of them belong to the exons of expressed genes, consistent with a previous study suggesting a positive correlation between methylation and exon expression level[27]. However, CpG methylation status at three heterochromatin sites is less informative. Methyl-RRBS signals at H1 indicate low levels of CpG methylation, but given H1 is located in a gene desert, the biological implication of this observation is unclear. For H3, we noticed moderate CpG methylation near genes that are not expressed (i.e., genes upstream of H3), in line with the transcription repression inside LAD.

Considering transcription factor CTCF is involved in mediating the insulator activity[28], we also examined H1-H3 for intrinsic CTCF binding sites that might interfere with our screening assay. CTCF ChIP-Seq data in HCT116 cells revealed no endogenous CTCF binding sequences within 30 kb distance to target sites H1-H3 (Fig. 1b), supporting that the barrier activity, if detected, would be contributed by DNA elements from the reporter plasmid integrated via SHIELD.

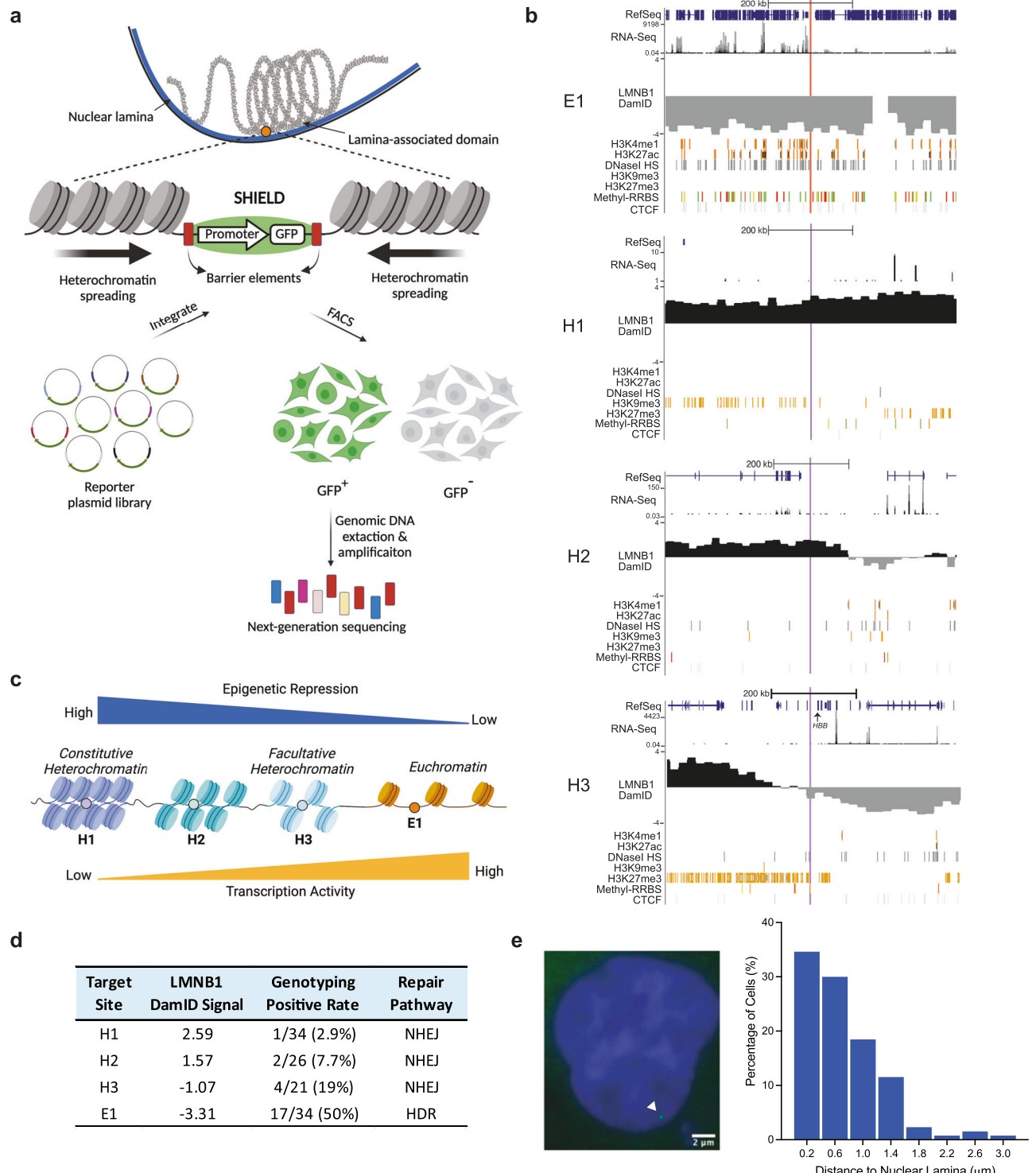

**Fig. 1 | Schematic overview of SHIELD and candidate loci with distinct epige-netic landscapes. a** Overview of SHIELD for high-throughput screening of barrier-type DNA elements. **b** The epigenetic landscape at four selected loci, including, from top to bottom in each figure, the RNA-seq track, the LMNB1 DamID signal, histone marks H3K4me1 and H3K27ac, DNaseI hypersensitive sites (HS), histone marks H3K9me3 and H3K27me3, Methylation-RRBS and CTCF ChIP-seq tacks in the selected HCT116 cell line. The source of each track can be found in Supplementary Table 3. For Methyl-RRBS track: red = 100% methylation, yellow = 50% methylation, green = 0% methylation. Vertical lines (red = E1, purple = H1, H2 or H3) denote the position of inserted landing pads and thus the reporter integration sites. **c** Schematic comparing the levels of epigenetic repression and transcription activity at four sites. **d** Summary of the CRISPR/Cas9-mediated KI of landing pad and the detected DSB repair pathways. Genotyping PCRs are listed in Supplementary Fig. 2. **e** Representative microscopic image of the H1 locus (Blue = DAPI, Green = GFP-TetR) and the distance distribution histogram of H1 to nuclear lamina (*N* = 130).

## Establishment of chassis cell lines by CRISPR/Cas9 and characterization of integrase activity at heterochromatin

We created HCT116 chassis cell lines by inserting the landing pad at each locus using CRISPR/Cas9 (Supplementary Fig. 1A). The linear DNA donor consisted of a blasticidin resistance gene (*BlaR*) driven by the strong EF1α core (EFS) promoter, and a PhiC31 integrase attP site was placed in between for reporter integration. Donor DNA sequences and corresponding homology arms can be found in Supplementary Table 2. For the H1 donor, we included an extra -1.5 kb 48-mer TetO array for imaging purpose[22]. From the genotyping polymerase chain reaction (PCR) of isolated clones (Supplementary Fig. 2), we observed significantly higher knock-in (KI) rate at E1 (17/34, 50%) compared to three heterochromatin loci H1-H3 (2.9–19%, Fig. 1d), consistent with previous studies[17,29]. Moreover, sequencing of KI junctions (Supplementary Fig. 3) revealed precise donor insertion at target E1, whereas small insertions and deletions (indels) were observed at both junctions from heterochromatin KI clones (H1#15, H2#24 and H3#16), indicating the error-prone non-homologous end joining (NHEJ) repair pathway was adopted. In line with previous reprots[22,29], our results suggest the chromatin structure dictates the outcome of Cas9-mediated KI, and NHEJ was the preferred repair pathway at heterochromatin. To validate the subnuclear localization of H1, we visualized the H1 locus using the clone H1#15 by SHACKTeR[22]. Indeed, we observed a peripheral localization of H1 (Fig. 1e) with a median distance of 0.6 μm (*n* = 130) to the nuclear periphery, suggesting inserting the landing pad did not significantly perturb the intranuclear localization of the target locus[22].

Previous works based on the landing pad mostly aimed to achieve active expression of integrated transgenes[21,30]. As a result, the scope of the integrase-based landing-pad strategy is currently limited to open chromatin, and its performance at compact heterochromatin has not been evaluated. Hence, we next set out to characterize the integrase activity at three heterochromatin sites (H1–H3) using E1 as a control. For this purpose, we used a 3.7 kb donor plasmid containing only the *PuroR* marker gene to select cells with on-target integration (Supplementary Fig. 4A).

We focused on three parameters: the overall integration rate, on-target integration rate, and integration fidelity. The overall integration rate was determined by the number of colonies surviving puromycin selection. Interestingly, we did not observe a significant difference in the colony forming units from four chassis cell lines (Fig. 2a, left), suggesting the chromatin structure may not significantly affect integrase activity. However, we noticed varying sizes of puromycin-resistant (PuroR) colonies (Fig. 2a, Supplementary Fig. 4B), with the majority of H1-PuroR colonies being much smaller than E1-PuroR clones. This size difference suggests a slower growth rate of H1-PuroR cells under the same pressure, likely caused by the repressive environment at H1 that suppressed *PuroR* expression. We next examined the on-target integration rate by junction genotyping PCRs. Intriguingly, all 53 clones analyzed were positive (Fig. 2a, Supplementary Fig. 4C), indicating a high on-target integration rate of integrase at both euchromatin and heterochromatin. To assess potential off-target integration at endogenous pseudo sites, we designed primers (Supplementary Table 5) targeting the top 3 pseudo sites previously identified from the human genome (Supplementary Fig. 5A)[31]. A similar PCR-based approach was also adopted to evaluate the off-target integration by a different integrase in mammalian cells[32]. From 10 clones with confirmed on-target integration, we observed no PCR amplicons corresponding to the off-target donor integration at any of the 3 pseudo sites (Supplementary Fig. 5B), indicating a strong preference of the integrase towards the bona fide attP site in the landing pad over endogenous pseudo sites.

Furthermore, sequencing of integration junctions revealed precise insertion of donor DNA by the integrase with no indels observed at all targets (Supplementary Fig. 4D), including heterochromatin sites where Cas9-based KI resulted in significant indels formation

(Supplementary Fig. 3B). Collectively, these results demonstrate the advantages of the large serine integrase over the CRISPR/Cas9 system for efficient insertion of plasmid-sized DNA (>3 kb) with high specificity and fidelity, particularly for targets at compact heterochromatin, thus establishing the foundation of SHIELD.

## Probing the silencing potential of selected loci

We estimated the relative epigenetic repression at each site by surveying various epigenetic marks (Fig. 1c). To test our hypothesis, we designed an *EGFP*-based reporter system (Fig. 2b) to probe the silencing potential of selected loci. Notably, the reporter cassette was placed in such an orientation that upon integration it would be in the opposite direction to the EFS promoter to minimize the potential effect EFS promoter may exert on *EGFP* expression. We selected three promoters of varying strength to drive *EGFP* expression, including the F9, SV40 (simian virus 40) and UBC (human ubiquitin C gene) promoter (Supplementary Table 6). To compare the strength of these promoters side by side, we integrated reporter plasmids carrying the F9-*EGFP*, SV40-*EGFP* or UBC-*EGFP* cassette at the housekeeping gene locus E1 and evaluated *EGFP* expression, which revealed F9, SV40 and UBC as the weak, intermediate and strong promoter, respectively (Fig. 2c).

Next, to probe the silencing potential of selected heterochromatin sites, we started with the F9-*EGFP* reporter since a weaker promoter would be more prone to epigenetic silencing, thus increasing the sensitivity of the assay[33]. Because of the high specificity of SHIELD with no detectable off-target integration (Supplementary Fig. 5B), we applied the PuroR polyclonal population directly for analysis without clonal isolation. Indeed, we noticed prominent chromosome position effects on *EGFP* expression with significant silencing observed at three heterochromatin sites H1–H3 (Fig. 2d). Importantly, H1–H3 exhibited distinct silencing kinetics, with silencing occurring the most rapidly at H1 but more slowly at H2 and H3 (Fig. 2d). By contrast, the F9-driven *EGFP* expression remained relatively stable at E1 (Fig. 2d). Collectively, these results demonstrate that: (i) the presence of the EFS promoter likely did not perturb the local repressive landscape at heterochromatin, and (ii) H1 was the most repressive loci, in line with our assumption. *EGFP* silencing at H3 was surprising as H3 is located at the LAD boundary (Fig. 1b). Nonetheless, it suggests LAD boundaries could be subject to silencing that is likely caused by heterochromatin spreading from inside the LAD. Furthermore, we observed promoter-dependent *EGFP* silencing at H1 with silencing occurring much faster under two promoters of viral origins (F9 and SV40) than the constitutive UBC promoter (Supplementary Fig. 6A), consistent with studies showing viral promoters are more susceptible to complete silencing in mammalian cells[34].

Intriguingly, due to the large dynamic ranges of two stronger promoters (SV40 and UBC), we were able to discern a silencing pattern (Fig. 2e) that was distinct from the previously reported *all-or-none* silencing phenomenon[35,36]. Specifically, besides the complete shutdown of *EGFP* (i.e., the emergence of EGFP⁻ population), we noticed a second EGFP⁺ population of lower intensity gradually emerged over time (Fig. 2e). This pattern was the most obvious in the H1_UBC-*EGFP* histogram where the dominant peak of the EGFP⁺ population gradually shifted from high intensity to low (Fig. 2e, right). This distinct silencing pattern prompted us to revisit the premises of SHIELD.

For a potential explanation (Supplementary Fig. 7), we took into account the recent discovery of gene repression at the mRNA level[37]. In fact, cellular EGFP intensity depends on both *EGFP* transcription and its mRNA translation. Zhou et al. recently discovered in human cells the rixosome is recruited by the polycomb repressive complexes to cleave newly synthesized mRNA at heterochromatin, preparing it for degradation[37]. Considering the heterochromatic nature of H1 and its location inside a highly repressive LAD (Fig. 1b), it is possible that the nascent EGFP mRNA at H1 is subject to

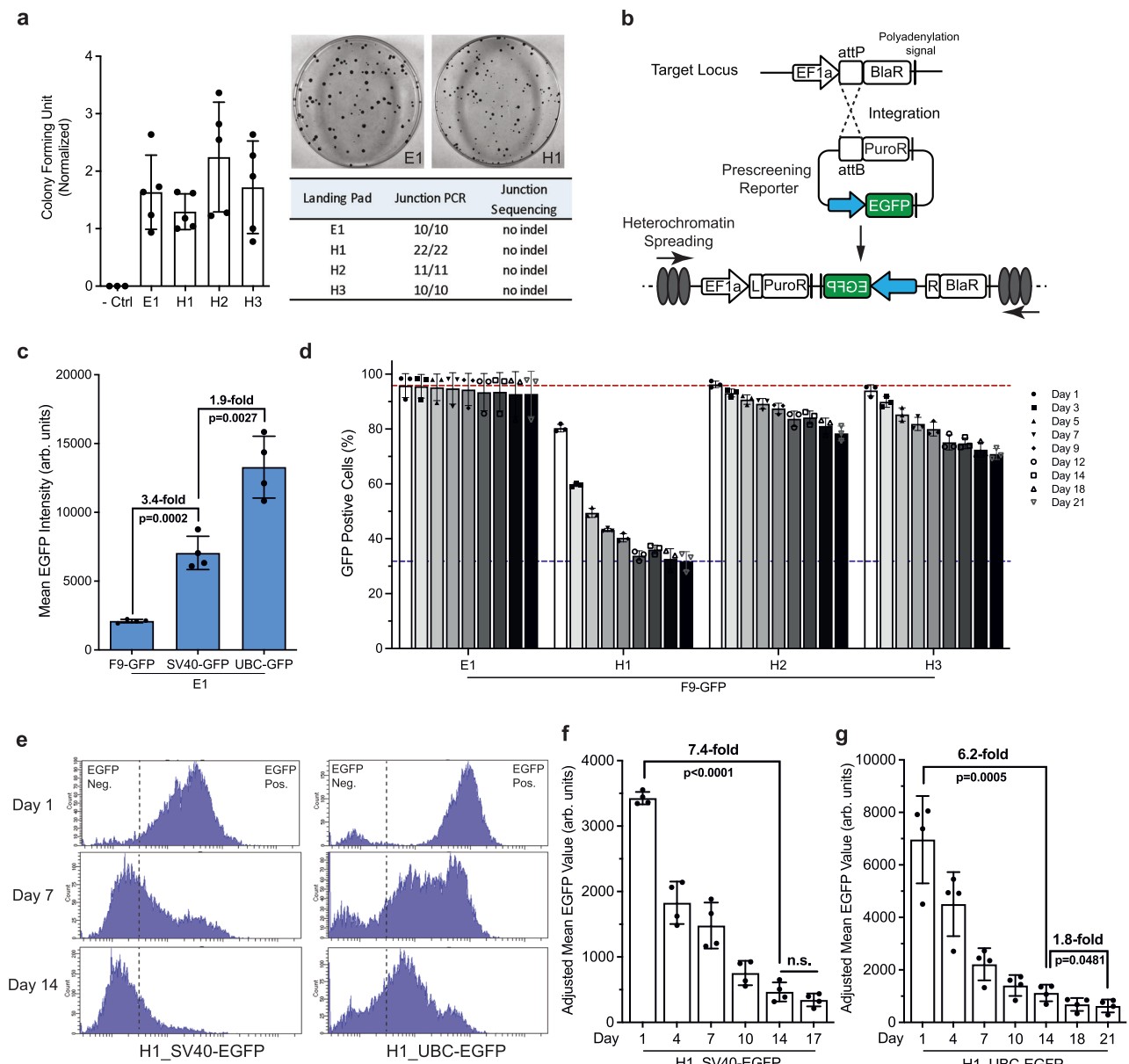

**Fig. 2 | Characterization of integrase activity and probing the silencing potential of selected heterochromatin sites. a** Left, colony formation assay following integrase-mediated reporter insertion at four target sites ($N = 5$ biological replicates) and negative control (i.e., no reporter plasmid, $N = 3$). Top right, representative images from E1 and H1 chassis cell line with PuroR colonies fixed in 100 mm culture plates after puromycin selection. Bottom right, genotyping positive rate and sequencing results of randomly isolated PuroR clones. **b** Design of the reporter system to probe the silencing potential of selected loci. Blue arrow denotes the promoter driving the reporter gene *EGFP*. L = attL, R = attR. **c** Promoter strength comparison of F9, SV40 and UBC. Reporter plasmids with different promoters were integrated at E1 to achieve a direct comparison of promoter strength at the same chromosomal context and in a single-copy manner. $N = 4$ biological replicates. Error bars represent means ± SD. *P* value was calculated by two-tailed unpaired *t*-test. **d** Percentage of EGFP+ population over time following puromycin

removal. The F9-*EGFP* reporter was integrated at four selected sites and *EGFP* expression was monitored over 21 days. Day 0: puromycin removal. $N = 3$ biological replicates. Error bars represent means ± SD. Red dashed line: EGFP+ cells percentage on day 1 with reporter integrated at E1. Blue dashed line: EGFP+ cells percentage on day 21 with reporter integrated at H1. **e** Representative histograms showing EGFP expression under two promoters (left: SV40, right: UBC) at H1 at indicated time points following puromycin removal. Dashed line denotes the boundary between EGFP- (autofluorescence) and EGFP+ populations. **f, g** EGFP expression level (mean fluorescence intensity) over time under either SV40 promoter (**f**) or UBC promoter (**g**) at H1. $N = 4$ biological replicates. Error bars represent means ± SD. Day 0: puromycin removal. The mean EGFP intensity was adjusted based on the florescence intensity of standard beads on day 1 to account for day-to-day fluctuations of the flow cytometer. *P* value was calculated by two-tailed unpaired *t*-test. n.s. = not significant ($p > 0.05$).

rixosome-triggered degradation, leading to the decrease in the mean EGFP intensity of the EGFP+ population (Supplementary Fig. 7). In this regard, we also included the mean fluorescence intensity (MFI) of the population, in addition to the percentage of EGFP+ cells (%), as another meaningful metric to reflect transgene repression at H1. Interestingly, when *EGFP* was driven by two strong promoters (SV40 and UBC) of distinct strength (Fig. 2c), we

observed a similar exponential decay in the population MFI over time (Fig. 2f, g).

To assess *EGFP* silencing in individual clones, we randomly isolated five colonies from the H1_SV40-*EGFP* polyclonal population and monitored *EGFP* expression in each isolated clone up to 40 days after puromycin removal. We observed a significant decrease in MFI from all five clones within the first 7 days (Supplementary Fig. 6B).

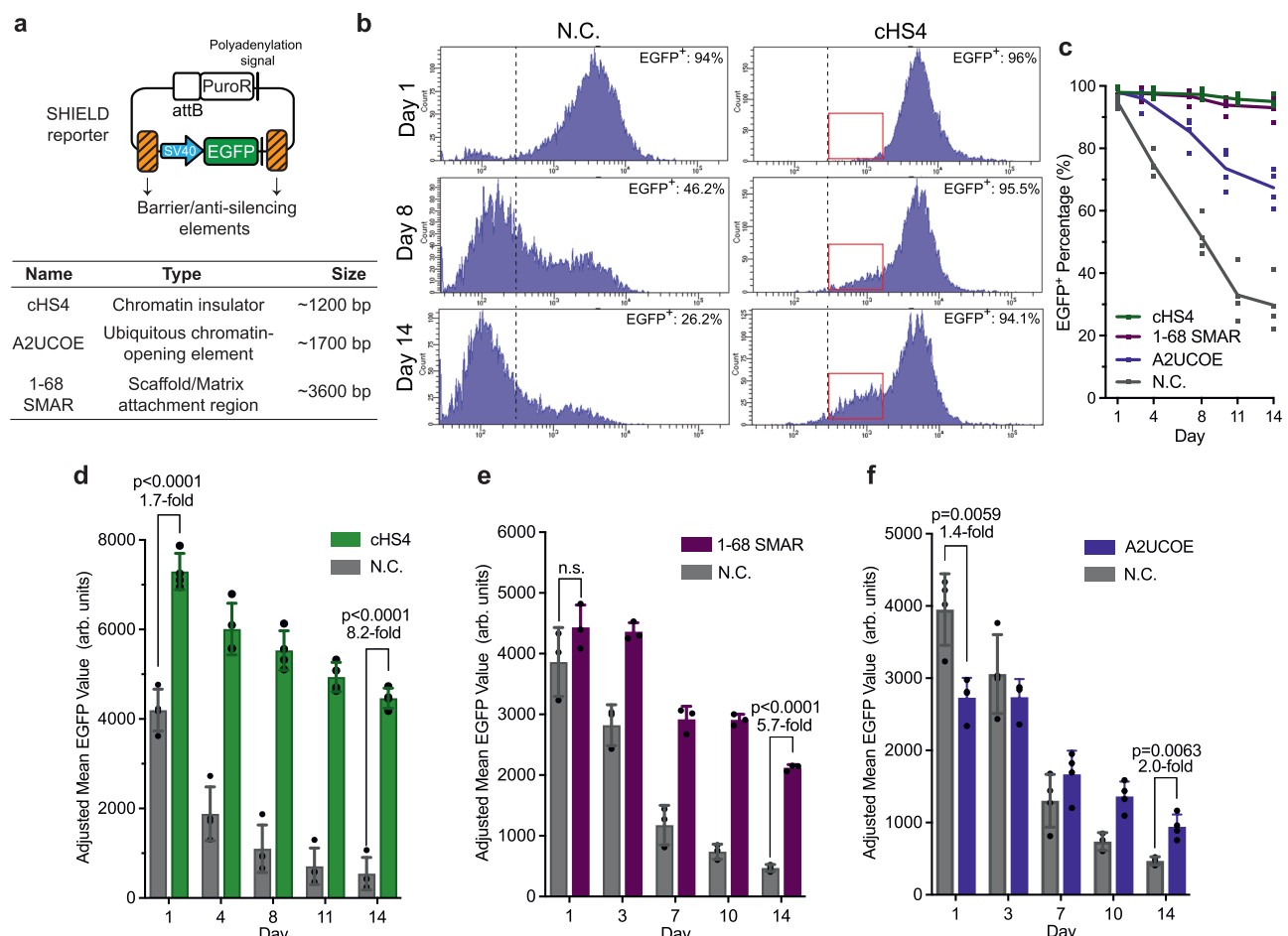

**Fig. 3 | SHIELD proof-of-principle with existing barrier/anti-silencing elements.** **a** Design of the reporter plasmid where the reporter cassette was flanked by selected barrier/anti-silencing elements, including full-length cHS4, A2UCOE and S/MAR 1−68. For each reporter, the upstream and downstream elements flanking the *EGFP* reporter gene were identical (see Methods for reporter cloning). **b** Representative histograms showing SV40-*EGFP* expression at H1 without (left, negative control = N.C.) or with the flanking full-length cHS4 insulator (right) at indicated time points following puromycin removal. Dashed line denotes the boundary between EGFP⁻ (autofluorescence) and EGFP⁺ populations. Red boxes highlight the low-intensity EGFP⁺ population. **c** Percentage of EGFP⁺ population over time where the SV40-*EGFP* reporter flanked with the corresponding elements or without any flanking element (N.C.) was integrated at H1. Each dot represents a biological replicate. $N = 4$ biological replicates for all samples. Solid line represents mean. **d**–**f** EGFP expression at H1 over time (determined by the mean EGFP intensity) where the SV40-*EGFP* reporter was flanked with the corresponding elements (**d** cHS4; **e** 1–68 SMAR; **f** A2UCOE) compared to the reporter without any flanking element (N.C.). $N = 4$, 3 and 4 biological replicates in **d**, **e** and **f**, respectively. The MFI was adjusted based on the florescence intensity of standard beads on day 1 to account for day-to-day fluctuations of the flow cytometer. Error bars represent means ± SD. *P* value was calculated by two-tailed unpaired *t*-test, n.s. = not significant ($p > 0.05$).

Interestingly, we also noticed significant clone-to-clone variation, with ~5-fold reduction in MFI from clone 4 and up to ~46-fold reduction from clone 2. Nonetheless, silencing profiles of individual clones on average closely resemble that of the polyclonal population (Fig. 2f), supporting that polyclonal cells generated from SHIELD can directly used for analysis.

**Known barrier and anti-silencing elements are active inside the LAD**

To demonstrate SHIELD can be adopted to discover unknown barrier elements, we next sought to test whether the epigenetic repression at H1 could be mitigated by existing barrier or anti-silencing elements. We selected three well characterized barrier/anti-silencing elements for this purpose, including the prototypic chromatin insulator cHS4, the scaffold or matrix attachment region (S/MAR) 1−68, and the ubiquitous chromatin opening element (UCOE) derived from the promoter region of human HNRPA2B1-CBX3 housekeeping genes (A2UCOE) (Fig. 3a, Supplementary Table 7)[38–40]. The cHS4 element has been well studied as a chromatin insulator with potent barrier activity mediated by transcription factors including CTCF, VEZF1, and USF[41].

Certain S/MAR elements are also classified as insulators[3] and can increase transgene expression at repressive chromatin by binding to transcription factors such as special (A + T)-rich binding protein 1 (SATB1), nuclear matrix protein 4 (NMP4) or CTCF[40]. UCOEs are commonly referred as "anti-silencing elements" instead of "chromatin insulators", and they can protect transgene(s) from epigenetic silencing and variegation in mammalian cells[42]. The selected A2UCOE contains two CTCF binding sites, which likely contribute to its reported anti-silencing activity[42]. However, the barrier/anti-silencing activities of these elements have not been compared side by side at the same chromosome context, especially inside highly repressive LADs.

Reasoning the heterochromatin could encroach from both directions, we placed the selected element both upstream and downstream of the reporter cassette (Fig. 3a). We continued with the SV40 promoter due to its relatively large dynamic range and reported compatibility with various elements[40,43,44], and focused on the H1 locus due to its highly repressive nature. Encouragingly, when the reporter was flanked by the full-length cHS4, we observed a stable EGFP⁺ population (~95%) that lasted over 14 days after puromycin removal, in striking contrast to the negative control (Fig. 3b, Supplementary

Fig. 8A). Similarly, S/MAR 1–68 also enabled *EGFP* to retain active expression (>90% EGFP⁺) over 14 days at H1, whereas A2UCOE was less capable of preventing *EGFP* from complete silencing at H1 with ~70% EGFP⁺ cells observed on day 14 (Fig. 3c). In addition to EGFP⁺ percentage (%), which is an ON/OFF binary classification of reporter expression, we also monitored the population MFI following puromycin removal. On day 14, we observed an 8.2-fold, 5.7-fold or 2.0-fold higher MFI when the reporter was flanked by the full-length cHS4 (Fig. 3d), S/MAR 1-68 (Fig. 3e) or A2UCOE (Fig. 3f), respectively, compared to the negative control (i.e., no flanking elements). Hence, when challenged at the same highly repressive context, the full-length cHS4 significantly outperformed the other two in protecting transgene from epigenetic silencing. In addition, the difference in activity also suggests that these elements likely employed different pathways to counteract gene silencing[45].

Interestingly, despite the reporter being flanked by cHS4, we found that a second EGFP⁺ population of lower intensity gradually emerged over time (Fig. 3b, red box), resulting in a significant decrease in MFI (Fig. 3d). This low-intensity population accounted for ~40% of the EGFP⁺ population on day 40 (Supplementary Fig. 8B), and could potentially be explained by EGFP mRNA degradation as proposed above (Supplementary Fig. 7). We also observed similar decrease in MFI in the case of S/MAR 1-68 (Fig. 3e) or A2UCOE (Fig. 3f), which was also contributed by the complete silencing of *EGFP*. Nonetheless, these results collectively demonstrate that active transgene expression can be achieved at a highly repressive LAD locus (e.g. H1) provided that the transgene is shielded by potent barrier elements such as cHS4, thus paving the way for discovering unknown barrier elements using SHIELD.

To examine if the increased expression of *EGFP* flanked by barrier elements was associated with spatial relocalization of the H1 locus in the nucleus, we visualized H1 in two populations. We found ~24% increase ($p = 0.03$) in the mean distance of H1 to the nuclear periphery when comparing the A2UCOE population to the negative control (Supplementary Fig. 9), suggesting the A2UCOE element could function by repositioning the integrated transgene away from the nuclear lamina, albeit to a lesser degree.

## Pilot screening of enhancer-blocking elements for barrier activity by SHIELD

Having validated certain exisiting barrier elements could protect *EGFP* from rapid silencing at H1, we next applied SHIELD to test elements with unknown barrier activity. We started by testing six previously reported enhancer-blocking elements individually (Fig. 4a, Supplementary Table 8) before adopting a pooled strategy. This step was necessary to assess the sensitivity of SHIELD as the barrier activity of a more compact element (~300 bp), if any, would likely be weaker than the elements tested above (>1.2 kb). These 6 elements are active enhancer-blocking elements previously identified from either high affinity CTCF-binding sites or mammalian-wide interspersed repeats (MIRs) in the human genome[11,13]. Despite being active enhancer-blocking elements, their barrier activities have not been assessed before.

We integrated six reporter plasmids (Fig. 4a) at H1 and compared *EGFP* expression on day 14 to both the negative control (i.e., no flanking elements) and the reporter flanked by full-length cHS4. Among six elements tested, only three (A2, A4, MIR2) exhibited significant barrier activity, with MIR2 being the most potent as determined by both EGFP⁺ percentage and population MFI on day 14 (Fig. 4b, c). These results demonstrated that enhancer-blocking and barrier functions are likely separable for a given element, and SHIELD is capable of distinguish these two.

We further analyzed the sequence feature of six elements, focusing on three transcription factors CTCF, VEZF1 and USF based on a previously proposed model[41]. Interestingly, we noticed that the most potent barrier element MIR2 contains the most binding motifs for VEZF1 and USF, whereas the inactive element E2 has no putative VEZF1 binding motif (e.g., GGGG) despite being highly active for CTCF binding (Supplementary Table 9). This observation suggests the potential role of VEZF1 in counteract epigenic silencing, in line with the proposed model[41]. Focusing on MIR2, we also designed synthetic DNA elements consisting of 1-mer, 2-mer or 3-mer MIR2 repeat arrays and evaluated their barrier activities at H1. We found the barrier activity of MIR2 was copy number-dependent (Supplementary Fig. 10A, B), with the artificial 3-mer MIR2 element (~1.1 kb) exhibiting comparable barrier activity to the full-length cHS4 element (~1.2 kb) as determined by the population MFI on day 14 (Supplementary Fig. 10B). This copy number dependency further indicates certain sequence features are likely associated with the barrier activity.

## High-throughput screening of endogenous DNA elements by SHIELD

We next expanded the library to 1000 elements to demonstrate the high-throughput potential of SHIELD. These candidates were selected from the non-coding regions of human genome, consisting of 450 CTCF-high affinity binding sequences, 50 CTCF-low affinity binding sequences, 30 MIR elements located within 5 kb to LADs boundaries (LAD-Bound), 420 randomly picked MIR elements and 50 randomly generated DNA sequences (Fig. 4d). We continued with CTCF-binding sequences and MIR elements mainly due to their compact size (<300 bp) and potential barrier activity (Fig. 4b, c). In particular, MIRs are relatively understudied compared to CTCF-binding sequences for chromatin insulation, and were proposed to act in a CTCF-independent manner[11]. Sequence analysis revealed distinct features of CTCF-binding and MIR elements, with CTCF-binding sequences on average having higher GC content (50.2–52.4%) than MIRs (38.7–43.0%) (Fig. 4e). CTCF-binding sequences also share a conserved core motif for CTCF binding, whereas MIR elements are more AT-rich with no conserved motif (bit score <0.05) (Fig. 4f). These distinct sequence features suggest these two types of elements may function in different pathways.

To facilitate the construction of reporter library for high-throughput screening (HTS), we modified the reporter backbone by placing the full-length cHS4 element downstream of the SV40-*EGFP* cassette and inserted varying elements upstream of the promoter (Fig. 4g, Supplementary Table 10). To examine the quality of constructed plasmid library, we first manually checked the cloning efficiency by colony PCR, which revealed a 90% (9/10) correct insertion rate (Supplementary Fig. 11A). NGS analysis of the pooled plasmids further revealed a 98.5% coverage of designed elements in the constructed library, with a relatively equal distribution of each element according to the cumulative fraction distribution curve (Supplementary Fig. 11B).

We transfected reporter plasmids as a pool, obtained cells with stable integration at H1 and then removed the pressure to induce epigenetic silencing (Fig. 4h). Cells were sorted based on EGFP levels at indicated time points, their genomic DNAs extracted for PCR with integration-specific primers (Supplementary Table 11, Supplementary Fig. 12) and amplicons were barcoded and sequenced with Illumina MiSeq (Fig. 4h). For quick visual evaluation, we first plotted NGS data as a heatmap (Supplementary Fig. 13A). We noticed distinct element distribution in sorted populations compared to the pDNA library, indicating certain levels of enrichment (or the lack thereof) for library elements following SHIELD-based screening. Specifically, we found that CTCF-family elements (ID = 1-500) were on average more enriched than MIR elements (ID = 501-950) in the Low-EGFP population (Day 3), but less enriched than MIRs in the High-EGFP population (Day 3) (Supplementary Fig. 13A). Moreover, as silencing accumulated, we also noticed changes in element enrichment over time. For instance, certain CTCF-family elements (e.g., ID = 350-470) became less

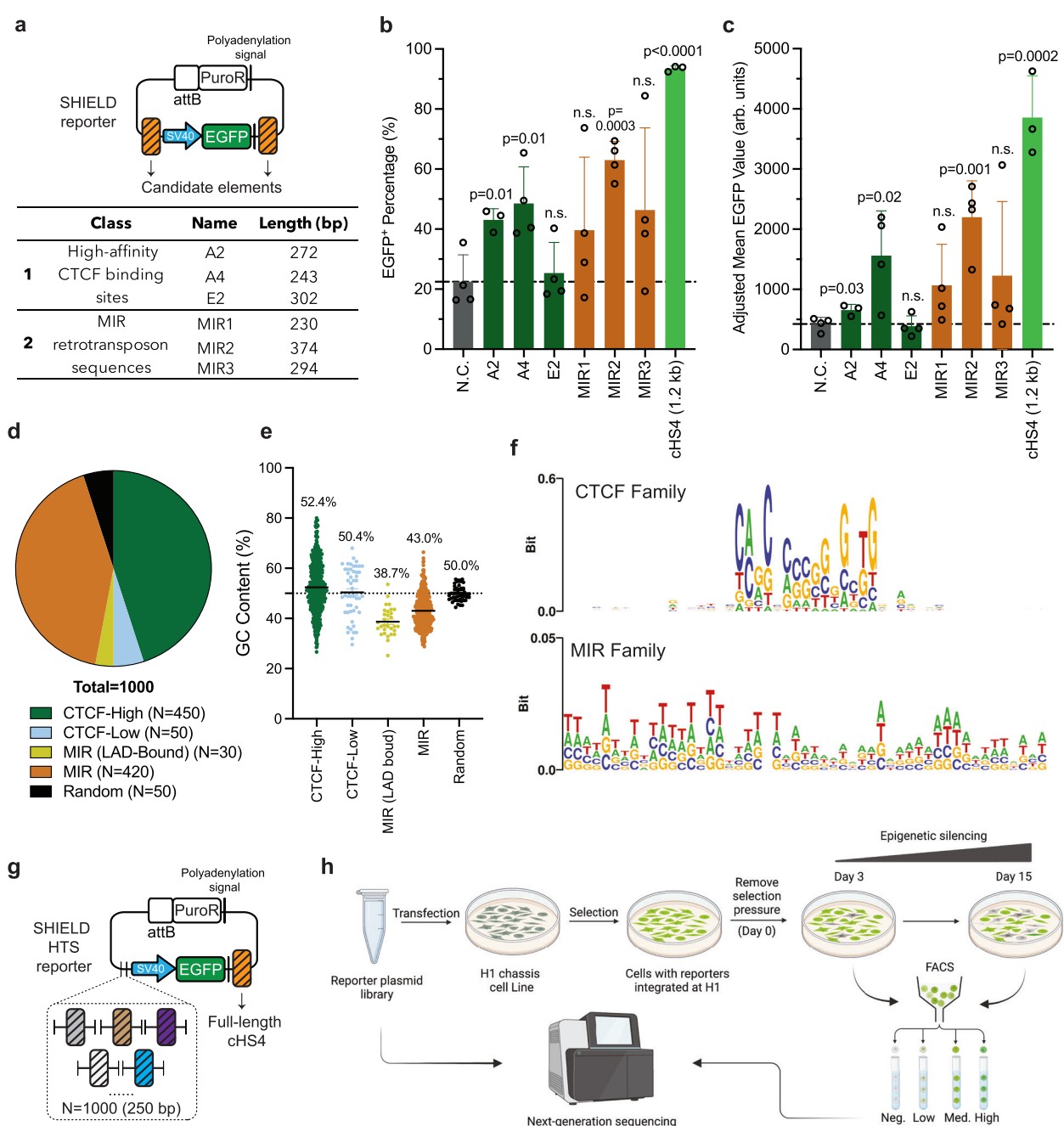

**Fig. 4 | Screening of six enhancer-blocking elements for barrier activity and the design of high-throughput screening library. a** Design of the reporter plasmid with SV40-*EGFP* flanked by indicated elements. For each reporter plasmid, the upstream and downstream flanking elements were identical. **b, c** Percentage of EGFP⁺ cells (**b**) or mean EGFP intensity (MFI) of the population (**c**) where the reporter was flanked with corresponding element or without any flanking element (N.C. = negative control) on day 14 after puromycin removal. *N* = 3 biological replicates for A2 and cHS4 (1.2 kb) or 4 biological replicates for all other samples. Error bars represent means ± SD. *P* value was calculated by two-tailed unpaired *t*-test, n.s. = not significant (*p* > 0.05). **d** Pie chart showing the composition of the high-throughput screening library. The library consists of 1000 DNA elements,

including 450 CTCF-binding sequences with predicted high affinity (named CTCF-High)[13], 50 CTCF-binding sequences with predicted low affinity (named CTCF-Low)[13], 30 MIR elements that are located within 5 kb to LAD boundaries (named MIR-LAD_bound)[11], 420 MIR elements randomly picked from the database[11], and 50 randomly generated DNA sequences. Each element is 300 bp in length, including 50 bp flanking sequences for cloning. **e** Dot plot comparing the GC content (%) of five subtypes in the library. Dashed line = 50%. **f** Sequence logo showing conserved motif(s) among CTCF family (top) and the MIR family (bottom). Note that the *y*-axis (Bit score) is plotted in different scale for better visualization: maximal score = 0.6 (for CTCF family) or 0.05 (for MIR family). **g** Design of the reporter for high-throughput screening. **h** Workflow of SHIELD-based high-throughput screening.

enriched in EGFP⁺ populations (Low/Med/High) sorted on day 15 compared to day 3 (Supplementary Fig. 13A), indicating they were less capable to establish a stable chromatin barrier.

Focusing on populations sorted on day 15, we plotted NGS results in volcano plots (Fig. 5a) for quantitative analysis. We found the majority of random sequences were significantly enriched (i.e., fold

change FC > 1.5, *p* < 0.05) in the EGFP-negative but not EGFP-positive populations (Fig. 5a), supporting SHIELD could filter out non-functional elements in a high-throughput manner. In addition, the best performing element MIR2 tested above (Fig. 4b,c) was captured as significantly enriched in both the EGFP-Med (FC = 4.3, *p* = 0.009) and EGFP-High population (FC = 2.0, *p* = 0.025) from HTS, further

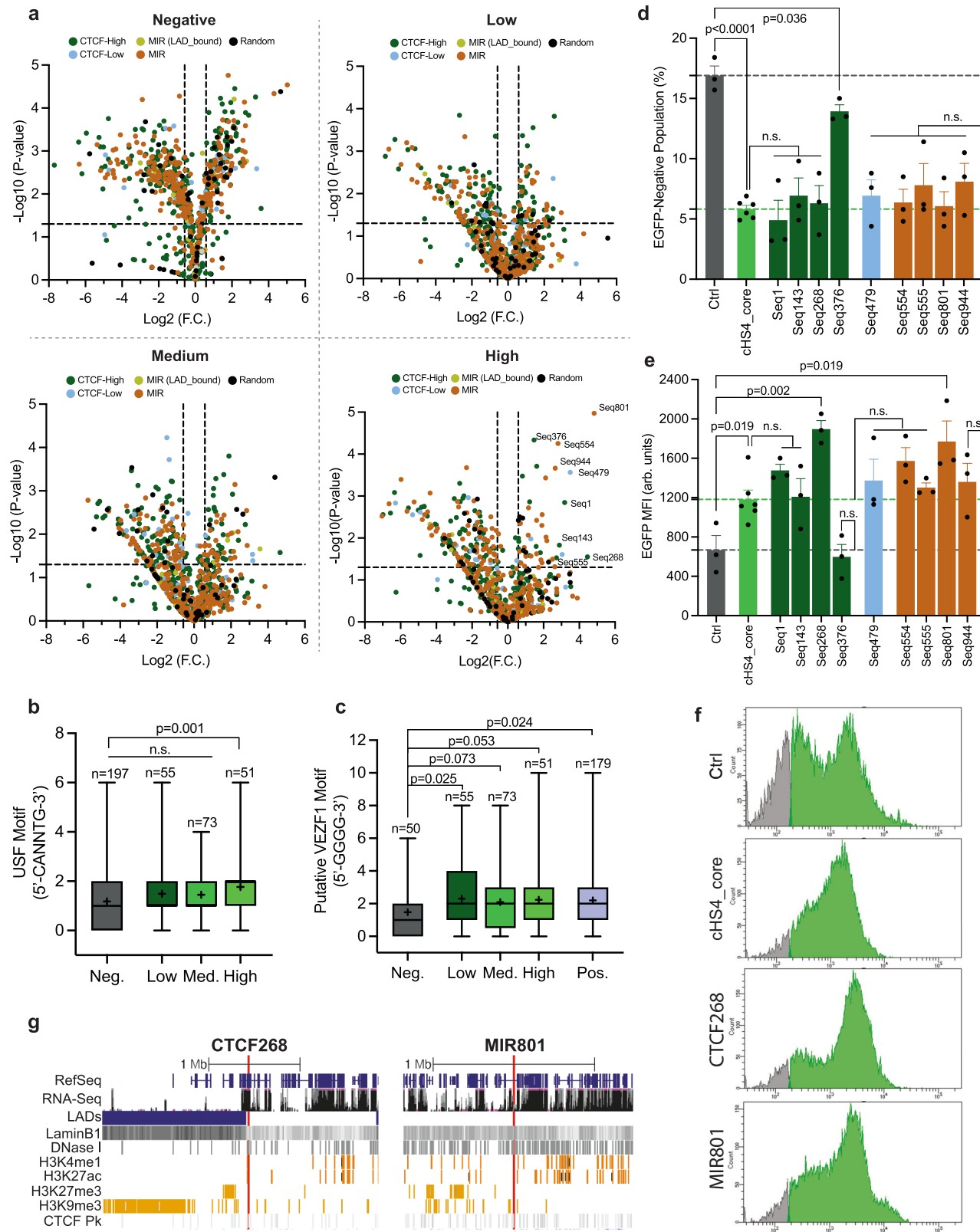

indicating the effectiveness of SHIELD. Among the 51 significantly enriched elements in the EGFP-High population, CTCF-high affinity elements accounted for 29% (Supplementary Fig. 13B), consistent with previous studies suggesting the important role of CTCF in establishing chromatin barrier[46]. Interestingly, however, the majority of elements enriched in the EGFP-High population were MIRs (53%) (Supplementary Fig. 13B), indicating MIRs may contribute to enhancing gene

expression in addition to establishing chromatin barrier. In addition, we found no significant difference in GC content of elements enriched in four populations with varying EGFP expression on day 15 (Supplementary Fig. 13C).

Taking advantage of our large dataset from HTS, we compared the prevalence of transcription factor USF and VEZF1 binding motifs in elements enriched in different populations. We found that in the

**Fig. 5 | High-throughput screening of barrier elements via SHIELD and validation. a** Volcano plots showing NGS results of four populations sorted on day 15: EGFP-Negative, EGFP-Low, EGFP-Medium and EGFP-High. FC = fold change. Horizontal dashed line: *P* value = 0.05. Vertical dash lines: FC = 1.5 (right) or 0.67 (left). *P* value was calculated by two-tailed unpaired *t*-test (*N* = 2). **b** Box plot comparing the number of putative USF binding motif (5′-CANNTG-3′) in elements significantly enriched in four populations sorted on day 15. Neg. = Negative. Med. = Medium. *P* value was calculated by two-tailed unpaired *t*-test. n.s. = not significant. Whiskers: min to max; Boxes: 25th to 75th percentiles; Middle line: median; "+": mean. **c** Box plot comparing the number of putative VEZF1 binding motif (5′-GGGG-3′) in elements significantly enriched in five populations sorted on day 15. For this comparison, we focused on the top 50 highly enriched elements in the EGFP-negative (Neg.) population. Pos. = EGFP-positive population (i.e., combination of Low, Medium and High). *P* value was calculated by two-tailed unpaired *t*-test. n.s. = not

significant (*p* > 0.05). Whiskers: min to max; Boxes: 25th to 75th percentiles; Middle line: median; "+": mean. **d**, **e** Validation of top hits from high-throughput screening. Data shown for EGFP-negative population percentage in (**d**) and median fluorescence intensity (MFI) in (**e**) on day 14 after puromycin removal. Reporter plasmids for validation were constructed as shown in Fig. 4g. Ctrl: no element. cHS4_core: the 5′ 250 bp of full-length cHS4. Gray dashed line: mean value of Ctrl. Green dashed line: mean value of cHS4_core. Error bars represent means ± SD. *N* = 6 for cHS4_core and *N* = 3 biological replicates for all other samples in (**d**) and (**e**). *P* value was calculated by two-tailed unpaired *t*-test. n.s. = not significant (*p* > 0.05). **f** Representative flow histograms of four populations on day 14. Gray: EGFP-negative. Green: EGFP-positive. CTCF268: seq268. MIR801: seq801. **g** Location and epigenetic information of two top hits identified from high-throughput screening: CTCF268 (seq268) and MIR801 (seq801).

EGFP-High population enriched elements contained significantly more USF binding sites than in the EGFP-Negative population (*p* = 0.001) (Fig. 5b). Similar pattern was also observed for VEZF1 when the top 50 enriched elements in the EGFP-Negative population were included for comparison (Fig. 5c). Collectively, these results are consistent with the pilot screening (Supplementary Table 9) and further support the original model based on the prototypical cHS4 insulator[41,47].

To validate HTS hits, we selected nine highly enriched elements in the EGFP-High population (indicated in the volcano plot) and included the cHS4 core region (250 bp) for comparison. We constructed ten reporter plasmids as shown in Fig. 4g, and transfected them separately into the H1 landing-pad cell line, together with the backbone plasmid (i.e., no upstream element) that served as a control (ctrl). We observed strong barrier activity from cHS4_core with ~5.8% EGFP⁻ population detected on day 14 compared to the ctrl (~16.9%) (Fig. 5d). Also, cHS4_core helped improve *EGFP* expression by ~2-fold at H1 on day 14 compared to the ctrl (Fig. 5e, f). Notably, eight out of nine (~89%) selected hits exhibited barrier activity that was comparable to the cHS4_core in terms of shielding *EGFP* from complete silencing (Fig. 5d). Furthermore, two elements, Seq268 (renamed CTCF268) and Seq801 (renamed MIR801), outperformed cHS4_core by further elevating EGFP expression at H1 as determined by the population MFI (Fig. 5e, f). Interestingly, we found CTCF268 is located close to a LAD boundary at the transition region between repressive and active chromatin domains, whereas MIR801 is located, although not close to a LAD boundary, within a region where repressive and active histone marks gradually switch (Fig. 5g, Supplementary Table 12). Their position in the genome indicates genomic context could be another factor that reflects barrier activity. Collectively, these results not only validated HTS outcome but demonstrated SHIELD could discover, to the best of our knowledge, previously unknown barrier elements with activities comparable to or better than the commonly adopted cHS4_core element in human cells.

## Discussion

In this work, we established SHIELD as a robust platform for high-throughput screening of barrier DNA elements in human cells. SHIELD is built upon the high specificity of serine integrase at heterochromatin and exploits the naturally occurring gene silencing inside LADs, thus enabling efficient screening of DNA elements under the same chromosome context in a systematic and high-throughput manner. Inserting exogenous DNA at heterochromatic regions (e.g., LADs) is technically challenging due to the compact chromatin structure. Hence, direct insertion of the reporter gene into repressive regions by Cas9 alone was deemed infeasible, especially when multiple insertions were needed in parallel for comparison. We discovered that PhiC31 integrase remained highly active and specific at three heterochromatin sites, a feature we took advantage of to establish SHIELD. The large serine integrase is mechanistically different from CRISPR/Cas9 in that it does not rely on endogenous DSB repair pathways for successful

DNA integration, which likely contributes to its retained activity and specificity at heterochromatic regions[48].

Previous studies including seminal works by Hathaway et al. and Bintu et al. reported silencing as all-or-none events in mammalian cells[36,49]. These studies, however, were limited by their adopted methods to artificially induce reporter silencing at transcriptionally active sites through targeted recruitment of transcription repressors such as KRAB. These approaches, although effective and efficient, were likely inadequate to fully recapitulate gene silencing in mammalian cells as they did not take into account the potential context dependency of gene silencing. For instance, the kinetics of artificially induced silencing at an otherwise transcriptionally active domain may be different from that of the naturally occurring gene silencing inside highly repressive LADs, as the latter may involve a concerted action of multiple silencing machineries that preferentially localize to repressive nuclear compartments such as the lamina. In this sense, our work complements previous studies by enabling the study of gene silencing in situ. SHIELD captured a silencing pattern that was distinct from the acknowledged all-or-none phenomenon, indicating that gene silencing in human cells likely employs diverse mechanisms beyond the all-or-none kinetics as previously described.

Epigenetic features that dictate transgene silencing rate at heterochromatin remain largely elusive in mammalian cells. DiPiazza et al. recently revealed that in fission yeast a critical density of H3K9me3 is required for heterochromatin propagation to enforce stable gene silencing[50]. Here in human HCT116 cells, by examining the silencing of the same reporter gene at epigenetically distinct endogenous loci that we intentionally selected with varying densities of H3K9me3 marks, we also observed distinct silencing kinetics that correlated well with not only H3K9me3 but also LMNB1 DamID level. In addition, Rival-Gervier et al. reported three classes of silencing (rapid, gradual or not silenced) following retrovirus integration in embryonic stem cells[51]. Our work provides direct evidence further supporting this kinetic classification of silencing in human cells (e.g., no silencing at E1, gradual silencing at H2 and H3, and rapid silencing at H1). More importantly, in the case of random integration by retrovirus, only a minority of clones (2/11, ~18%) were rapidly silenced, whereas the majority remained not silenced (6/11, ~55%)[51]. By contrast, SHIELD enabled efficient generation of clones where silencing is not only dominant but rapid, thus offering an advantageous platform for silencing-related studies by shortening the timeframe and simplifying the workflow. Interestingly, we noticed the LAD-induced rapid silencing at H1 closely resembles that of the artificially induced gene repression by CRISPR-based epigenome editors[52,53]. This similarity suggests the repressive landscape inside LADs may be locally reconstituted at open chromatin by programmable epigenome repressors such as CRISPRoff[52].

Our work also calls for more caution to distinguish the barrier activity from the enhancer-blocking activity of an identified DNA insulator. We tested the barrier activity of three CTCF-binding

sequences (A2, A4 and E2) previously identified as highly potent enhancer-blocking elements[13]. These compact elements (~300 bp) were shown to be much more potent (>6-fold) than the full-length cHS4 element (1.2 kb) by the enhancer-blocking assay. However, our SHIELD-based assay revealed that these elements exhibited only weak to no barrier activity compared to the full-length cHS4 (Fig. 4b, c), highlighting the discrepancy between these two properties for a given DNA insulator. This information is also important for synthetic biologists because it clarifies that these elements (A2, A4, E2), although attractive due to compact sizes, may not be suitable to replace the full-length cHS4 element when the goal is to minimize epigenetic silencing. For example, due to the lack of this information, a recent study adopted the A2 element with the intention to block epigenetic silencing, yet significant silencing of integrated transcription units was still observed in HEK293T cells[54].

Through SHIELD we performed, to the best of our knowledge, the first large-scale and high-throughput screening ($n = 1000$) of endogenous DNA elements for barrier activities. Proteins involved in establishing chromatin barriers in mammalian cells remain largely elusive, although CTCF is frequently posited as a key player. In this regard, there appears to be at least two types of barrier elements depending on the involvement of CTCF. MIR elements were proposed to function in a CTCF-independent manner[11], yet 6% (27/450) selected MIRs exhibited strong barrier activities. In comparison, 3% (15/500) of selected high-affinity CTCF-binding sequences were enriched in the same EGFP-high population. Hence, diverse mechanisms are likely employed to form heterochromatin boundary in addition to CTCF-mediated chromatin looping. In this view, transcription factors USF and VEZF1 are likely important contributors by recruiting active histone marks to resist the propagation of repressive histone modifications and by DNA demethylation, respectively. SHIELD identified two top hits (CTCF268 and MIR801) that outperformed the cHS4_core in improving transgene expression at H1 by 50-60%, and the endogenous nature of these two elements may render them less immunogenic than cHS4_core (chicken origin) for applications in human cells.

We note that our library elements were relatively short (i.e., 250 bp) due to the current synthesis limit for oligo pools. Hence, only MIRs and CTCF-binding elements were included for screening mostly due to due to their compact size, which inevitably limited the scope of current work. Nonetheless, considering various elements (e.g., cHS4, S/MAR, UCOE) remained functional at H1, as well as the large-cargo capability of the integrase (e.g., reporter plasmid carrying S/MAR 1–68 is >10 kb in size), future work would benefit from advanced DNA synthesis technologies to extend the length of synthetic DNA, thus further expanding the library scope.

It is possible to adapt SHIELD for studies that aim to elucidate silencing mechanisms or evaluate anti-silencing strategies in human cells. Recent development of genome editing techniques could further improve the workflow of SHIELD[55,56]. We anticipate our platform will also enable more thorough investigations of LADs as the "dark matter" in mammalian genomes[57].

## Methods

### Epigenetic information of selected loci
Chromosome coordinates of four selected loci are summarized in Supplementary Table 1. Ten tracks of each locus are provided to represent the epigenetic landscape in Fig. 1b, including the NCBI RefSeq, HCT116 RNA-Seq, HCT116 LMNB1 DamID, HCT116 H3K4me1 and H3K27ac, HCT116 DNaseI HS, HCT116 H3K9me3 and H3K27me3, DNA Methyl-RRBS and HCT116 CTCF ChIP-Seq. All tracks are publicly accessible at the UCSC Genome Browser except the HCT116 LMNB1 DamID signal track, which was created by the Bas van Steensel group as part of the 4D Nucleome project[24]. A summary of the sources of epigenetic information can be found in Supplementary Table 3.

### Cell culture and establishment of landing-pad cell lines by CRISPR/Cas9
HCT116 cells (ATCC #CCL-247) were cultured in McCoy's 5A medium (without phenol red, UIUC Cell Media Facility) supplemented with 10% tetracycline-free fetal bovine serum (FBS, Sigma-Aldrich). Cells were grown at 37 °C in a humidified 5% $CO_2$ incubator and routinely passaged following ATCC guidelines. To create chassis cell lines by CRISPR/Cas9, 0.5 million HCT116 cells were electroporated with 3 μg Cas9 protein (Integrated DNA Technologies, IDT), 2 μg single guide RNA (sgRNA) and 10 μg DNA donor using the Amaxa Nucleofector II device (Lonza) and Nucleofector kit V (Lonza). The sgRNAs were prepared using the GeneArt Precision gRNA Synthesis Kit (Thermo Fisher Scientific). DNA donors were amplified by polymerase chain reaction (PCR) from corresponding plasmid templates carrying either the EFS-attP-BlaR (for E1, H2, H3) or 48merTetO-EFS-attP-BlaR cassette (for H1) (Supplementary Table 2). PCR was performed with Q5 High Fidelity DNA polymerase (New England Biolabs, NEB), and product was purified with QIAquick PCR purification kit (Qiagen) following manufacturer's protocol. The 5′ ends of donor DNA were chemically labeled with PEG10 following a previously described protocol[58]. Two days after nucleofection, cells were plated onto multiple 100 mm plates (with 10 μg/ml blasticidin) with serial dilution for drug selection and colony isolation. Culture media (with 10 μg/ml blasticidin) was refreshed regularly during selection. After ~10 days, single colonies were picked into 24-well plate for expansion. Genomic DNA was extracted from each clone using QuickExtract DNA Extract Solution (Epicentre) after cells reached ~80% confluency. Genotyping PCR was performed with isolated genomic DNA and indicated primers (Supplementary Table 1) using Q5 High Fidelity DNA polymerase (NEB) following manufacturer's protocol. To verify the integrity of inserted donor, gel-purified junction PCR products (Qiagen Gel Extraction Kit) of selected clones for each target site were analyzed by Sanger DNA sequencing (ACGT Inc.).

### Characterization of integrase performance at selected loci
A 3.6 kb plasmid (Supplementary Fig. 4a) carrying the attB-PuroR cassette was used for this test. Briefly, 24 h before transfection, ~10^5 cells of each chassis cell line were plated per well in a 12-well plate. Transfection was performed the next day as follows: 500 ng pCAG-Integrase plasmid and 500 ng pattB-PuroR plasmid were mixed in 50 μl Opti-MEM media (Thermo Fisher Scientific), followed by the addition of 3 μl FuGENE HD reagent (Promega). The mixture was incubated at room temperature for 5 min before adding to each well and culture media was refreshed 1 day after transfection. Colony formation assay was performed to compare the overall integration rate at four loci. Briefly, 2 days after transfection, cells were split onto 100 mm plates (with 0.5 μg/ml puromycin) at 1:4 ratio to initiate drug selection. After ~10 days of puromycin treatment, each plate was washed with phosphate-buffered saline (PBS) and stained with crystal violet solution. Stained colonies were manually counted as shown in Supplementary Fig. 4b. To examine the on-target integration efficiency as well as integration fidelity, single colonies were isolated from 100 mm plates as described above. Genomic DNA was extracted from each clone after expansion and genotyping were performed as described above with primers listed in Supplementary Table 4. Out of 53 clones that were positive for junction PCRs, we analyzed the junctions of 10 clones (one for E1, three each for H1–H3) by Sanger DNA sequencing (ACGT Inc.). To assess potential off-target integration at endogenous pseudo sites, we performed genotyping PCRs (Supplementary Fig. 5) with primers listed in Supplementary Table 5.

### Visualization of H1 and image analysis
To visualize H1 inside the nucleus, a previously described imaging system was used[22]. Briefly, cells were transduced with lentivirus (F9-TetR-GFP-IRES-PuroR) at low multiplicity of infection. Transduced cells

were later fixed with 4% paraformaldehyde and stained with DAPI for imaging. Fixed samples were analyzed with a Personal DeltaVision deconvolution microscope equipped with the 60X oil objective (NA 1.4) and the CoolSNAP HQ slow-scan CCD camera (Roper Scientific, Vianen, Netherlands). The Z-stack images were captured with a step size of 0.2 μm, and the images were deconvoluted using the Softworx program (GE Healthcare, Little Chalfont, UK). Image analysis was performed using the ImageJ software (National Institutes of Health) to measure the distance of H1 to the nuclear periphery in Fig. 1e and Supplementary Fig. 9. To do this, we selected the Z section where the nucleus was in focus and the fluorescent signal of H1 was relatively high, and then used the straight line tool in ImageJ to measure the distance from the center of H1 to the nearest nuclear boundary, which was determined by DAPI.

### Reporter cloning for probing silencing potential and SHIELD pilot screening

Plasmid pattB-*PuroR* was used as a backbone to create *EGFP* reporter plasmids. To construct reporter plasmids with *EGFP* under different promoters (Supplementary Table 6), the corresponding F9/SV40/UBC-*EGFP*-polyA cassette was inserted between MluI and MfeI restriction sites in the backbone by dual digestion and ligation. Plasmid pattB-*PuroR*-SV40-*EGFP* was then used as a backbone to construct reporter plasmids with flanking DNA elements. Such cloning was done by either Gibson Assembly or two rounds of restriction digestion and ligation. The cHS4 element was amplified from pC-HBH (Addgene #88896). The A2UCOE and S/MAR 1-68 were also amplified from HCT116 genomic DNA, and the expected sequences were listed in Supplementary Table 7 together with cHS4. Six elements in Fig. 4a were also amplified from HCT116 genomic DNA and their corresponding chromosome coordinates are summarized in Supplementary Table 8. The MIR2 repeat array was assembled by Golden Gate Assembly (NEB) into a helper plasmid pUC19 (Addgene #50005), and the assembled array was then inserted into pattB-*PuroR*-SV40-*EGFP* plasmid as described above.

### Integration of reporter plasmids and flow cytometry analysis

One day before transfection, ~$10^5$ cells of the chassis clone were plated per well into a 12-well plate. Transfection was performed the next day as follows: 500 ng pCAG-Integrase plasmid and 500 ng EGFP reporter plasmid were mixed in 50 μl Opti-MEM media (Thermo Fisher Scientific), followed by the addition of 3 μl FuGENE HD reagent (Promega). The mixture was incubated at room temperature for 5 min before adding to each well and media was refreshed 24 h after transfection. Depending on confluency, cells were split at 48 h or 72 h post transfection with 1:2 splitting ratio into a 6-well plate (with 0.5 μg/ml puromycin) to initiate selection. After ~9 days of selection, cells were transferred into a 12-well plate with 1:2 or 1:3 splitting ratio (depending on confluency) and puromycin was removed from media. EGFP expression was analyzed by flow cytometry starting the next day as day 1, and was monitored over time at indicated time points.

For HTS validation, transfection was performed in 24-well format to increase throughput. Briefly, 250 ng pCAG-Integrase plasmid and 250 ng EGFP reporter plasmid were mixed in 50 μl Opti-MEM media, followed by the addition of 1.5 μl FuGENE HD reagent. Media was refreshed 24 h after transfection, and puromycin selection was initiated 48 h after transfection and lasted for ~8 days. Cells were then collected for flow cytometry.

For flow cytometry, cells were collected and resuspended in PBS. Samples were analyzed on a BD LSR Fortessa flow cytometer (BD Bioscience) and at least 10,000 events were recorded per sample. Gating strategies for flow cytometry can be found in Supplementary Fig. 14. To account for potential day-to-day variation of the cytometer, standard fluorescent beads (Flow-Check Fluorospheres, Beckman Coulter) were analyzed in each run, and the mean EGFP signal of the beads was used to adjust EGFP signal of samples analyzed after day 1 (Eq. 1):

$$\text{Adjusted Sample Mean EGFP}_{\text{day}n} = \text{Sample Mean EGFP}_{\text{day}n} \times \frac{\text{Beads Mean EGFP}_{\text{day}n}}{\text{Beads Mea EGFP}_{\text{day}1}}$$

(1)

### HTS library design and construction

The CTCF binding sequences and MIR elements were chosen from previously published databases[11,13]. For CTCF-binding sites we selected the top 450 elements (CTCF-High) and the bottom 50 elements (CTCF-Low). We included CTCF-low affinity binding sites to evaluate the contribution of CTCF-binding to barrier activity. For MIR elements, since no ranking system was applied in the original database, we first filtered out MIRs that were located close to LAD boundaries (defined as <5 kb in distance for simplicity). For this purpose we downloaded the coordinates of previously identified constitutive LADs[23] and performed intersection analysis between MIR and LAD boundaries using BEDTools[59], which yielded 30 such MIRs (named MIR_LAD-Bound) including the MIR2 element tested in pilot screening. The other 420 MIRs were randomly selected from the database. To streamline the cloning process, we adjusted the length of each selected element to 250 bp. We later found that 146 elements were adjusted to 225 bp by mistake, but this minor difference did not affect library cloning and data analysis. We also included 50 random DNA sequences (250 bp each) generated by an online software named Random DNA Sequence Generator. Sequences with Esp3I recognition sites were excluded as they would interfere with library cloning.

We added 50 bp flanking sequences to each element for reporter cloning, and obtained an oligo pool (*N* = 1000, Supplementary Data 1) synthesized by Twist Bioscience. The oligo pool was then PCR amplified with the Fwd and Rev primers shown in Supplementary Table 10 using the KAPA HiFi HotStart ReadyMix (Roche) at the following thermal cycling conditions: 95 °C for 3 min, (98 °C for 20 s, 69 °C for 15 s, 72 °C for 15 s) for 15 cycles, 72 °C for 1 min, then held at 4 °C. PCR products were analyzed in 1.5% agarose gel and the target bands were extracted and purified. 5 ng of the gel-purified PCR products were assembled with 100 ng HTS reporter plasmid backbone in a 20 μl Golden Gate assembly reaction at the following thermal cycling conditions: 37 °C for 5 min, (37 °C for 5 min, 16 °C for 10 min) for 35 cycles, 16 °C for 30 min, 37 °C for 45 min, 80 °C for 5 min, then held at 4 °C. The reaction was then treated with Plasmid-Safe DNase (Lucigen) per manufacture's protocol. 1 μl of the reaction was transformed into 25 μl of NEB 10-beta electrocompetent E. coli cells (NEB C3020K). We performed two transformation in parallel following a previously established protocol[60], and estimated the total number of colony forming units to be $4.72 \times 10^5$ and $8.42 \times 10^5$ for each transformation, which represents a 472-fold and 842-fold coverage of the SHIELD plasmid library (*N* = 1000). Plasmids were extracted using a Qiagen Plasmid Maxi Kit per manufacturer's protocol. HTS library plasmid DNA (pDNA) quality was further determined by NGS (Supplementary Fig. 11b).

### HTS library transfection and FACS

For library transfection, ~300,000 H1 chassis cells were plated per well in a six-well plate at 24 h before transfection. We performed 24 transfections in total to ensure the reproducibility of our screening outcome. Briefly, we calculated the absolute integration efficiency as ~0.4% based on the colony formation assay in Fig. 2a. Hence, 24 transfections, each with ~600,000 cells at the time of transfection (assume 24-h doubling time), would yield roughly ~57,600 clones, which represents > 50-fold coverage of the SHIELD library during the actual screening pipeline. This fold-of-coverage is significantly higher

than a previous study that also adopted the integrase-based high-throughput screening (25-fold coverage)[61].

For each transfection, 1 μg of pCAG-PhiC31 integrase plasmid and 1 μg of the purified library plasmid pool (diluted in 100 μl Opti-MEM) were mixed with 6 μl of FuGene HD (Promega) and added to each well. 1 day after transfection, cells were split into new wells to initiate puromycin selection. Puromycin selection (0.5 μg/ml) was performed for ~9 days until no significant cell death was observed. After selection, cells from 24 transfections were pooled and around 1.5 million cells were plated per plate onto two 100 mm cell culture plates, which served as two biological replicates for epigenetic silencing at H1. Flow cytometry of pooled cells immediately following puromycin removal showed >96% EGFP$^+$ population, indicating the high efficiency of reporter integration and puromycin selection. FACS was performed on day 3 and day 15 after puromycin removal using a Thermo Fisher Bigfoot Spectral Cell Sorter. FACS gate settings are shown in Supplementary Fig. 12.

### NGS sample preparation, data processing and analysis
To prepare sample for NGS, genomic DNA of each population was extracted using PureLink Genomic DNA Mini Kit (Invitrogen #182001) and ~ 200 ng were used as the template for PCR amplification (25 μl) with primers listed in Supplementary Table 11. For NGS of reporter plasmid library, pDNA from Maxi Prep was used as template. PCR products were purified either with magnetic beads or through gel electrophoresis, and used as templates for the second-stage index PCR with Nextera barcoded primers. Index PCR products were then purified again with magnetic beads or through gel electrophoresis, quantified with Qubit Fluorometer (Thermo Fisher Scientific), and mixed in equal molar ratio for NGS. Sequencing was performed at the UIUC Roy J. Carver Biotechnology Center DNA Services lab using the Illumina MiSeq system with the 2 × 250 nt capacity.

Fastq files were generated and demultiplexed with the bcl2fastq v2.20 Conversion Software (Illumina), and then evaluated for quality control (QC) with FastQC. Fastq files were then converted to Fasta files using the FASTX-Toolkit. Fasta files were trimmed to remove shared linker sequence and to retain regions in each read with high quality (Phred Score > 25.75 as determined by FastQC). To map each NGS read to the library, we created a local blast database containing SHIELD library sequences, and used the BLAST + 2.7.1. module to perform alignment analysis of NGS data against library database. For post-alignment data processing, we first removed duplicates (i.e., one read being mapped to multiple library sequences) and kept only one alignment with the highest bitscore (lowest e-vaule) for each read. In addition, we removed alignments with more than 3 mismatches, or more than 3 gaps, or with a total mapped length less than the maximum possible minus 10 nucleotides. We then counted the frequency of each library sequence in the processed blast data file, and determined the relative abundance (%) of each library sequence in the sorted population by dividing its counts with the total counts of all 1000 sequences in the corresponding NGS sample.

For the heatmap shown in Fig. 5a, the abundance of each element in the sorted EGFP$^+$ populations represents the average of two biological replicates, whereas in the pDNA library it represents the average of two technical replicates (i.e., the forward and reverse NGS reads of the pDNA sample). For volcano plots, the fold change and associated $p$ value for each element was calculated against the pDNA library. Three volcano plots (EGFP-Low/Med/High) were created with data obtained from two biological replicates. Unfortunately, one of the EGFP-Negative samples sorted on day 15 was contaminated during sorting, so the EGFP-Negative volcano plot was plotted based on two technical replicates (i.e., the forward and reverse NGS reads of the uncontaminated sample). Nonetheless, we obtained an average Pearson correlation value $R = 0.80$ between the remaining biological replicates ($R = 0.79$, 0.83 and 0.77 for the EGFP-Low, Medium and High populations sorted on day 15, respectively), consistent with previously reported high reproducibility of landing-pad based screening pipeline.

### Statistics and reproducibility
All quantitative data are presented as mean ± standard deviation (SD), with the exception of Fig. 3c where the solid lines represent the mean value of replicates. Statistical methods are summarized in the figure legends. Statistical analyses were performed with GraphPad Prism 9.

### Reporting summary
Further information on research design is available in the Nature Portfolio Reporting Summary linked to this article.

## Data availability
All data generated or analyzed during this study are included in the main text or supplementary information. Plasmids used in this study will be deposited to Addgene and are available from the corresponding author upon request. NGS data can be accessed at GEO with the accession number GSE236198. Source data are provided with this paper.

## Code availability
We used standard FASTX-Toolkit and BLAST+ 2.7.1 module to process and analysis NGS sequencing data. Parameters used can be found in the Methods section under "NGS sample preparation, data processing and analysis". The code used for NGS data analysis is publicly available at GitHub (https://github.com/mzhang100/SHIELD-NGS) and Zenodo (https://doi.org/10.5281/zenodo.8288219 [https://zenodo.org/record/8288219]).

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

## Acknowledgements

This work was supported by the U.S. National Institutes of Health (1U54DK107965 and 1UM1HG009402 to H.Z. and UM1HG011593 to D.M.G.). We thank Prof. Andrew Belmont and Dr. Ipek Tasan for helpful discussions. We thank Prof. Michele Carlos (Stanford Genetics) for sharing reagents and helpful discussions about the integrase system. We thank Dr. Yunan Luo for help with library design. We thank Dr. Mayandi Sivaguru (Cytometry and Microscopy to Omics Facility, UIUC) for assistance with FACS. We thank Christopher J. Fields (High-performance Biological Computing, UIUC) and Dr. Qiqi Tian (Zhao lab, UIUC) for help with NGS data analysis. We thank Dr. Sandra Kay McMasters for providing cell culture media (UIUC, SCS Cell Media Facility). Light Microscopy was performed at the Light Microscopy Facility in the Department of Molecular and Cellular Biology (UIUC). We also acknowledge the Bas van Steensel lab at the Netherlands Cancer Institute for generating the genome-wide LMNB1 DamID data in HCT116 cells as part of the 4D Nucleome Project. Certain figures were created using Biorender.

## Author contributions

M.Z. conceived the idea. M.Z. and H.Z. designed the research. D.G. provided plasmids with the integrase and the landing-pad cassette. M.Z. performed the experiments with help from M.E. and S.M. M.Z. analyzed NGS data with help from A.G.B. H.Z. supervised the research. M.Z. and H.Z. wrote the manuscript with input from all authors.

## Competing interests

The authors declare no competing interests.
