## [Peer Review File · Nature Communications]

Reviewers' Comments:

Reviewer #1:

Remarks to the Author:

Following on from the original work of Felsenfeld, it is known that the genome is punctuated with elements that can act either to insulate genes from the surrounding influence of positive and negative environmental effects (barrier insulators) or interrupt enhancer-promoter interactions (enhancer blockers). In experimental systems, both types of element protect transgenes from position effects resulting from the chromosomal environment in which a transgene integrates. While the role of CTCF bound elements acting as enhancer blockers has been intensively investigated, their role as barrier insulators has not been studied to the same extent. In this paper, the authors explore this question: How do different DNA elements protect a transgene from silencing when it is introduced into a repressive heterochromatic environment. This is an important outstanding question and if answered would contribute to our understanding of how certain elements can form a barrier to the spread of heterochromatin.

To address this Zhang et al developed an assay, SHIELD, to functionally screen insulators that block the spreading of heterochromatin. To that end, they create an integrase-based landing pad in pre-selected lamina associated domains (H1, H2 and H3) LADs in the human HCT116 cell line. At each site they integrate a reporter gene tagged with EGFP, and flanked by test barrier elements. EGFP is only detected if the reporter is effectively shielded by the barrier elements from the heterochromatic environment. Sequences of effective barrier elements are then revealed by NGS of DNA from sorted EGFP expressing cells. Using this approach, they identified two novel high affinity binding CTCF sequences with substantial barrier activity and describe a potential role for VEZF1 transcription factor in preventing silencing.

Major points

Although the three test beds (H1, H2 and H3) all classify as LADs they are clearly different forms of heterochromatin. Otherwise, these landing areas are poorly characterised. The only repressive mark assayed is H3K9me3 while there are many others that should have been evaluated, not least H3K27me3, HP1, nuclease/transposase accessibility, DNA Methylation etc. In addition, they have not tested these areas for intrinsic CTCF binding sites which could also influence the results. In summary, the test sites are poorly characterised and it would help their interpretation of the proposed assay to do so.

The second issue concerns the nature of the sequences they initially test. Although CTCF sites are well established insulators, matrix attachment regions (MARs) and Ubiquitous Chromatin Opening Elements (UCOEs) are relatively poorly defined elements and at least to my understanding are not generally accepted as defined classes of insulators. Despite this, all such elements seem to have some activity in their assay. The major problem with the experiments as they stand is that these elements range in size from 1215-3613bp, so it is completely unclear as to which sequences are acting as insulators. The authors do not provide any information on this and do not even mention whether the MARs and UCOEs contain CTCF binding sites.

The third point concerns analysis of the experimental data. Figure 3F illustrates the most relevant findings, but how were these data acquired, analysed and normalised? Why are some of the sequences very highly present in all of the populations/groups (68, 55), whereas others are completely absent (22,23)? This could most easily be explained by a technical bias rather than a biological phenomenon, and means these results are impossible to interpret. It cannot be explained by cells simply being negative for GFP (these cells are not sorted and sequenced), because Supplementary Figure 9A shows that this population is very small at day 3, and yet there are many sequences absent from day 3. This could be explained either by integration efficiency or biases introduced during the PCR amplification step. It is crucial to control for these biases.

Point 4 concerns some of the mechanistic conclusions of their findings. The authors claim they can distinguish between transcription and translation dependent silencing in their assay. Although it is true that both these processes affect the cellular EGFP levels, it is not clear how the authors can distinguish between the two. Just the level and spread of EGFP levels can be due to many factors, such as the half-life of the protein which means some fluorescence can still be detected if the silencing of the locus has just started. The authors note that ribosome-triggered degradation might play a role, but this is pure speculation. It would be interesting to mention in the discussion as a possibility, but does not belong in the main body of the text or Figure 2D. If they want to

claim any ribosome involvement they have to perform a ChIP for ribosome subunits (TEX10 or MDN1) and show that it's present at their integration site, preferably in higher levels after more days of puromycin withdrawal than in the beginning. They could also show that nascent transcription is still present but protein levels are down, if they want to claim regulation on a translational/RNA stability level. The authors have made no attempt to evaluate changes in the chromatin or DNA methylation in silenced or non-silenced integrants to determine how these elements are being silenced or protected from silencing.

The final screen involves only ~100 sequences which is far too small a group from which to draw conclusions. Highlighting the role of VEZF based on previous conclusions from the Felsenfeld lab is speculative and they provide no experimental evidence for a role for VEZF.

In summary, the aim of this work to clarify how barrier insulators may be identified and classified is laudable and points out a missing area of current research. Clearly identifying and designing better insulators for experimental and therapeutic purposes would be of considerable value. Unfortunately, the approach presented includes only preliminary and inconclusive results.

Reviewer #2:

Remarks to the Author:

Summary

In this manuscript, the authors introduce a new platform, called SHIELD (Site-specific heterochromatin insertion of elements at lamina-associated domains) for the high-throughput screening of barrier-type DNA elements in human cells by exploiting the naturally occurring heterochromatin spreading inside lamina-associated domains (LADs). The authors hypothesized that selecting LAD locus sites with a strong repressive epigenetic landscape could rapidly silence a reporter gene, which when shielded by potent barrier elements, would retain active expression. To study this they chose sites susceptible to epigenetic silencing to conduct their experiments. This manuscript addresses an important challenge in the field of transgene silencing by identifying barrier DNA elements that can be used to mitigate this challenge.

Comments for authors:

1. On page 4, line 83 the authors state that using PhiC31 integrase circumvents the need for tedious clonal selection. I think the authors need to be careful with this statement because utilizing the PhiC31 integrase system doesn't prevent random integration, it only increases efficiency of integration into the docking site. Scientists will still need to confirm the correct integration at the docking site location and that it was not randomly integrated into the genome.
2. The observation that a second EGFP+ population of lower intensity gradually emerges in 14 days is interesting. Have the authors considered taking the study out to 21 days or even longer to see what this population does?
3. The assertion that VEZF1 is recruited to the artificial sequences is not explained and there is no data supporting this claim (page 13 line 290).
4. The authors have identified low, medium, and high expression levels (Supp Fig. 9), have any studies been done to further separate and analyze each population for epigenetic markers/molecules that may be impacting expression levels?
5. Figures 2E, Supp. Figs. 9, and 10 need a label on the x-axis

Reviewer #3:

Remarks to the Author:

In this report Zhang et al. describe SHIELD, an assay for screening functional barrier DNA elements in the human genome. The significance of the question being addressed is clearly defined and the authors describe an interesting screening strategy for profiling the activity of putative barrier DNA elements. Unfortunately, there are substantial technical deficiencies that preclude publication of the manuscript in its current form.

With the authors having placed the emphasis of the manuscript on the development of a

Reviewer #1

Following on from the original work of Felsenfeld, it is known that the genome is punctuated with elements that can act either to insulate genes from the surrounding influence of positive and negative environmental effects (barrier insulators) or interrupt enhancer-promoter interactions (enhancer blockers). In experimental systems, both types of element protect transgenes from position effects resulting from the chromosomal environment in which a transgene integrates. While the role of CTCF bound elements acting as enhancer blockers has been intensively investigated, their role as barrier insulators has not been studied to the same extent. In this paper, the authors explore this question: How do different DNA elements protect a transgene from silencing when it is introduced into a repressive heterochromatic environment. This is an important outstanding question and if answered would contribute to our understanding of how certain elements can form a barrier to the spread of heterochromatin.

Response: We thank the reviewer for summarizing the background and reiterating the significance of our work. This comment is insightful and echoes with our motivation. We also added some of Felsenfeld's previous studies as references to provide more background of our work.

To address this Zhang et al developed an assay, SHIELD, to functionally screen insulators that block the spreading of heterochromatin. To that end, they create an integrase-based landing pad in pre-selected lamina associated domains (H1, H2 and H3) LADs in the human HCT116 cell line. At each site they integrate a reporter gene tagged with EGFP, and flanked by test barrier elements. EGFP is only detected if the reporter is effectively shielded by the barrier elements from the heterochromatic environment. Sequences of effective barrier elements are then revealed by NGS of DNA from sorted EGFP expressing cells. Using this approach, they identified two novel high affinity binding CTCF sequences with substantial barrier activity and describe a potential role for VEZF1 transcription factor in preventing silencing.

Response: We appreciate the reviewer's time and efforts in reviewing our work. We agree with the major points raised by the reviewer and have significantly revised our manuscript to address these issues. Detailed responses are summarized below.

Major points

1) Although the three test beds (H1, H2 and H3) all classify as LADs they are clearly different forms of heterochromatin. Otherwise, these landing areas are poorly characterised. The only repressive mark assayed is H3K9me3 while there are many others that should have been evaluated, not least H3K27me3, HP1, nuclease/transposase accessibility, DNA Methylation etc. In addition, they have not tested these areas for intrinsic CTCF binding sites which could also influence the results. In summary, the test sites are poorly characterised and it would help their interpretation of the proposed assay to do so.

Response: We thank the reviewer for pointing out the lack of characterization of three heterochromatin test beds. We agree with this comment and have included more epigenetic information to address this issue. Specifically, we included the H3K27me3 mark, DNaseI HS (hypersensitive site), DNA methylation-RRBS (reduced representation bisulfite sequencing) and CTCF ChIP-Seq in the HCT116 cell line around four target sites (Figure 1B). A detailed evaluation of these additional epigenetic marks indeed revealed H1, H2 and H3 as heterochromatin of distinct features, as described below:

1) H3K27me3: a repressive histone modification that often correlates with facultative heterochromatin. We noticed significantly enriched H3K27me3 marks near heterochromatin site H3, but not at H1 or H2. This observation suggests that H3 is likely part of facultative heterochromatin as opposed to the constitutive heterochromatin where H1 is located (e.g., enrichment of H3K9me3). In fact, we found that H3 is located upstream of the human β -globin gene (*HBB*), a gene that is inactive in several cell types but actively transcribed in HCT116 as evidenced by the RNA-Seq track. This observation is also in line with the dynamic nature of facultative chromatin where genes are either silenced or actively transcribed depending on the cell type.

2) DNaseI HS: regions of chromatin that are sensitive to cleavage by the DNase I enzyme. Overall, within the ~400 kb window analyzed, we found more DNaseI HSs at H2 and H3 than H1, indicating a less condensed chromatin structure at H2 and H3 than at H1. This is consistent with the fact that H3 is located upstream of an actively transcribed gene *HBB* in HCT116, which would require a relatively open chromatin structure for gene expression.

3) DNA methylation-RRBS: endogenous DNA methylation. We found abundant CpG islands around E1 but much fewer at three heterochromatin sites H1, H2 and H3. This difference in CpG density is likely due to the difference in gene density between euchromatin and heterochromatin, as CpG islands are typically associated with promoters of genes. Specifically, the majority of CpG dinucleotides near E1 are not methylated (green), indicating an active transcription environment. Although the remaining CpG dinucleotides at E1 appear moderately (orange) or highly methylated (red), most of them belong to the exons of expressed genes, consistent with a previous study suggesting a positive correlation between methylation and exon expression level. However, CpG methylation status at three heterochromatin sites is less informative. Methyl-RRBS signals at H1 indicate low levels of CpG methylation, but given H1 is located in a gene desert, the biological implication of this observation is unclear. For H3, we noticed moderate CpG methylation near genes that are silent (i.e., genes upstream of H3), in line with the transcription repression inside LAD.

4) CTCF ChIP-Seq: intrinsic binding sequences for the transcription factor CTCF. We agree with the reviewer’s comment that intrinsic CTCF binding sites at/near the test beds might interfere with our screening, and we thank the reviewer for pointing out this issue. We examined H1, H2 and H3 for intrinsic CTCF binding sites by analyzing the CTCF ChIP-Seq data in HCT116 cells. Fortunately, we found no endogenous CTCF binding sequences within 30 kb distance to target sites H1-H3, suggesting that the CTCF-related barrier activity, if detected, would be contributed by DNA elements from the reporter plasmid integrated via SHIELD.

We did not include heterochromatin protein 1 (HP1) ChIP-Seq as the reviewer suggested because we were not able to find a published database for this information in HCT116 cells, and performing an independent HP1 ChIP-Seq is considered beyond the scope of our work. Nonetheless, considering HP1 typically associates with H3K9me3 marks, the H3K9me3 track could serve as a good indicator of HP1 binding sites near four target sites.

In summary, we included additional epigenetic information as described above, which reveals that H3 is likely part of facultative heterochromatin whereas H1 exhibits typical constitutive heterochromatin features. In comparison, H2 appears in-between on the heterochromatin “spectrum”. Based on these additional analysis, we added an extra figure (see below, as shown in Figure 1C) showing the relative chromatin compactness, transcription activity, and putative epigenetic repression level of four selected sites for a direct comparison.

Schematic comparing the levels of epigenetic repression and transcription activity at four sites.

2) The second issue concerns the nature of the sequences they initially test. Although CTCF sites are well established insulators, matrix attachment regions (MARs) and Ubiquitous Chromatin Opening Elements

(UCOEs) are relatively poorly defined elements and at least to my understanding are not generally accepted as defined classes of insulators. Despite this, all such elements seem to have some activity in their assay. The major problem with the experiments as they stand is that these elements range in size from 1215-3613bp, so it is completely unclear as to which sequences are acting as insulators. The authors do not provide any information on this and do not even mention whether the MARs and UCOEs contain CTCF binding sites.

Response: We thank the reviewer for this comment. We agree that S/MARs and UCOEs are relatively poorly defined and less studied compared to CTCF-based insulators such as cHS4. However, certain S/MARs located near the boundaries of active chromatin domains exhibit barrier activity and are classified as barrier-type insulators (Felsenfeld et al., 2002, PMID: 11825869). Another study also included S/MARs as genomic insulators alongside cHS4 (Goetze et al., 2005, PMID: 15743822). In addition, it was shown the prototypical insulator cHS4 is also associated with the nuclear matrix (Yusufzai and Felsenfeld, 2004, PMID: 15169959), suggesting a potential overlap in mechanism between cHS4 and S/MARs.

The S/MAR selected in our work, S/MAR 1-68, could counteract the variegation of transgene expression potentially by adjusting DNA curvature and nucleosome positioning, as well as recruiting transcription factors like SatB1, NMP4 and Hox-like family proteins (Girod et al., 2007, PMID: 17676049). However, CTCF was not listed as a core transcription factor with binding sites in S/MAR 1-68 (Girod et al., 2007), and we did not notice significant CTCF-binding signal when examining the CTCF ChIP-Seq data in HCT116 cells (image below). Hence, S/MAR 1-68 likely functions in a CTCF-independent manner.

RNA-seq and CTCF ChIP-Seq at S/MAR 1-68 element (orange) in HCT116 cells

We included the A2UCOE in our work because it has recently emerged as a potent epigenetic regulator that prevents the variegation of transgene expression and silencing in mammalian cells (Pfaff et al., 2013, PMID: 23307570; Ackermann et al., 2014, PMID: 24290698; Müller-Kuller et al., 2015, PMID: 25605798). The A2UCOE is a methylation-free CpG island from the promoter region of human HNRPA2B1-CBX3 housekeeping genes and can counteract transgene silencing through promoter demethylation, recruiting active histone marks and preventing the propagation of repressive histone marks. There are two CTCF binding sites experimentally determined in A2UCOE (Müller-Kuller et al., 2015, PMID: 25605798), which were also observed in the CTCF ChIP-Seq data in HCT116 cells (image below). Hence, CTCF likely contributes to the barrier activity of A2UCOE.

RNA-seq and CTCF ChIP-Seq at A2UCOE (blue) in HCT116 cells

In summary, we chose these three elements from distinct families with the goal to compare their activities under the same highly repressive chromosome context (i.e., inside the LAD) side by side. Such a comparison has not been performed before due to technical infeasibility, but could be achieved with our SHIELD platform. We note that currently there is a lack of terminology that properly defines or classifies these elements. As a result, they are sometimes referred to as chromatin modifying elements, epigenetic regulators, or simply “anti-silencing” elements. Nonetheless, the lack of proper terminology does not affect

the quality of our experiment results, which clearly suggest the full-length cHS4 as the best performing element in preventing epigenetic silencing at a highly repressive LAD locus (Figure 3C-F). We also included the following sentences in the main text for better clarification:

“The cHS4 element has been well studied as a chromatin insulator with potent barrier activity mediated by transcription factors including CTCF, VEZF1 and USF. Certain SMAR elements are also classified as insulators and can increase transgene expression at repressive chromatin by binding to transcription factors such as special (A+T)-rich binding protein 1 (SATB1), nuclear matrix protein 4 (NMP4) or CTCF. UCOEs are commonly referred as “anti-silencing elements” instead of “chromatin insulators”, and they can protect transgene(s) from epigenetic silencing and variegation in mammalian cells. The selected A2UCOE contains two CTCF binding sites, which likely contribute to its reported anti-silencing activity. However, the barrier/anti-silencing activities of these elements have not been compared side by side at the same chromosome context, especially inside highly repressive LADs.”

3) The third point concerns analysis of the experimental data. Figure 3F illustrates the most relevant findings, but how were these data acquired, analysed and normalised? Why are some of the sequences very highly present in all of the populations/groups (68, 55), whereas others are completely absent (22,23)? This could most easily be explained by a technical bias rather than a biological phenomenon, and means these results are impossible to interpret. It cannot be explained by cells simply being negative for GFP (these cells are not sorted and sequenced), because Supplementary Figure 9A shows that this population is very small at day 3, and yet there are many sequences absent from day 3. This could be explained either by integration efficiency or biases introduced during the PCR amplification step. It is crucial to control for these biases.

Response: We thank the reviewer’s comment on the technical details of our screening process and data analysis. To fully address this concern as well as the reviewer’s final comment on the small size of our previous library (n~100), we completely re-designed our library and expanded it to 1000 DNA elements (Figure 4D), which represents a 10-fold increases in the library size. A detailed description and comparison of elements in the new library can be found in our revised manuscript (Figure 4D-F). Briefly, the new library consists of 500 CTCF-binding sequences (including 450 high-affinity and 50 low-affinity binding sites), 450 MIR elements (including 30 MIRs located within 5 kb to LAD boundaries and 420 randomly picked MIRs) and 50 randomly generated DNA sequences for control purpose.

We reconstructed the reporter plasmids library (pDNA) and repeated high-throughput screening with the revised workflow shown in Figure 4H. We adopted three additional approaches to better control for potential technical biases that may affect the screening outcome. First, we determined the composition of the pDNA library by NGS, which revealed a relatively equal distribution of elements (Supplementary Figure 11B and 13A) with a coverage of 98.5% of the selected 1000 elements. Secondly, immediately following the removal of selection pressure, we split the polyclonal cells into two separate populations (day 0), which served as two “biological replicates” that underwent LAD-induced silencing and cell sorting independently from each other. The results from these two biological replicates were then used to generate the volcano plots shown in Figure 5A. Thirdly, instead of simply focusing on the absolute read abundance (%) of each element in sorted populations, we calculated their relative fold change (F.C.) and corresponding *p* value by comparing against their abundance in the original pDNA library. Each element was assigned with two parameters (F.C. and *p* value) which were used to create the volcano plots shown in Figure 5A, a commonly adopted method to interpret data from large-scale and/or high-throughput screening experiments (e.g., Bao et al., 2018, PMID: 29734295).

By adopting these additional approaches as well as a more stringent analysis of NGS results (see Methods section for NGS sample preparation, data processing and analysis), we believe our data quality has been significantly improved. For instance, by visually examining the NGS heatmap (Supplementary Figure 13A) we no longer noticed elements that were highly present in all populations regardless of EGFP-expressing level. Instead, for most library elements we found a distribution pattern that was dependent on EGFP expression level. The image below shows one of many such examples. Despite having a relatively high

abundance in the pDNA library, element A was present with high abundance only in the EGFP-medium population sorted on day 3 and was further enriched in this population sorted on day 15, with low abundance in EGFP-high populations (day 3 and day 15). Element B had higher abundance in EGFP-low populations than the EGFP-high populations sorted on both days. Element C, by contrast, was mostly enriched in EGFP-high populations.

Abundance of certain library elements in the plasmid DNA library (pDNA) and sorted populations

Furthermore, we found that the vast majority of elements were present in at least one population sorted on day 3 (Supplementary Figure 13A) except the ones that were not present in the initial pDNA library (coverage = 98.5%). In summary, we believe we have taken necessary and effective approaches to address the reviewer's concerns on the technical fitness of our high-throughput screening assay. We identified two novel elements with barrier activities better than the *chs4_core* element following the screening of our newly designed library (Figure 5D-G). More detailed information can be found in our revised manuscript.

4) Point 4 concerns some of the mechanistic conclusions of their findings. The authors claim they can distinguish between transcription and translation dependent silencing in their assay. Although it is true that both these processes affect the cellular EGFP levels, it is not clear how the authors can distinguish between the two. Just the level and spread of EGFP levels can be due to many factors, such as the half-life of the protein which means some fluorescence can still be detected if the silencing of the locus has just started. The authors note that rixosome-triggered degradation might play a role, but this is pure speculation. It would be interesting to mention in the discussion as a possibility, but does not belong in the main body of the text or Figure 2D. If they want to claim any rixosome involvement they have to perform a ChIP for rixosome subunits (TEX10 or MDN1) and show that it's present at their integration site, preferably in higher levels after more days of puromycin withdrawal than in the beginning. They could also show that nascent transcription is still present but protein levels are down, if they want to claim regulation on a translational/RNA stability level. The authors have made no attempt to evaluate changes in the chromatin or DNA methylation in silenced or non-silenced integrants to determine how these elements are being silenced or protected from silencing.

Response: We thank the reviewer for this detailed comment, and we appreciate his/her suggestions on the experimental design. To explain the silencing pattern that was distinct from the all-or-none silencing phenomenon as we observed in our work, we proposed a model that took into account gene silencing as the translation level. Based on this model, transcriptional silencing is reflected by the percentage of EGFP-negative population (i.e., complete EGFP silencing), whereas translational silencing is indicated by the decrease in the MFI of the EGFP-positive population. Hence, we suggested in our original submission that SHIELD could distinguish transgene silencing at these two levels. We agree with the reviewer that the half-life of EGFP might influence the fluorescence intensity detected at a given time point. However, throughout our study, we intentionally set the time interval between two flow cytometry analyses to be at least 3 days (72 hrs) apart, which is significantly longer than the estimated half-life of GFP (~24 hrs). Therefore, the difference in EGFP intensity detected at two time points (e.g., Day 1 vs. Day 4, or Day 1 vs. Day 7) should be predominantly caused by epigenetic silencing.

We agree that the proposed model was partially based on a recent study (Zhou et al., 2022, PMID: 35355014), and our work did not directly confirm the involvement of rixosome during epigenetic silencing at H1. Nevertheless, we believe this model provides the most reasonable explanation for the observed silencing pattern. For scientific rigor, we toned down our claim in the revised manuscript and moved the corresponding figure from the main body to the supplementary information (Supplementary Figure 7).

As to the reviewer's last comment on efforts to evaluate changes in chromatin or DNA methylation status, we would like to point out that the focus of our work was to develop and characterize the SHIELD platform and demonstrate its potential for high-throughput screening. To the best of our knowledge, such a platform has not been established before particularly in human cells. Additional studies are warranted to reveal the mechanisms by which active elements protect the reporter from epigenetic silencing at the highly repressive LAD locus H1, but in our opinion they are beyond the scope of our current work. On this front, Majocchi et al. (PMID: 24071586) studied the mode of action of different epigenetic regulators including cHS4, S/MARs and UCOEs for their ability to prevent transgene silencing at artificial telomeres, and revealed epigenetic signatures associated with the reduced silencing including increased acetylation marks and decreased deposition of repressive marks on nearby nucleosomes. We cited this study in our revised manuscript as it provides relevant information considering both LADs and telomeres are highly repressive domains in human genome.

The final screen involves only ~100 sequences which far too small a group from which to draw conclusions. Highlighting the role of VEZF based on previous conclusions from the Felsenfeld lab is speculative and they provide no experimental evidence for a role for VEZF.

Response: We thank the reviewer for this comment and we agree that our previous library was relatively small in size. As we described in our response to the reviewer's third point, we expanded our library by an order of magnitude ($n = 1,000$) and reperformed our screening assay with the new library. Although it is possible to further increase the library size as the integrase-based screening could accommodate large library of variants ($n = 12,000$ demonstrated by Cao et al., 2021, PMID: 34230498), we limited our library to 1000 elements mainly due to the limited choices of candidate elements. Currently, there is limited knowledge of what constitutes a functional barrier element, and nor are there sufficient barrier elements with validated activities that could be adopted to train a computational model. As a result, we relied on two published databases (genome-wide CTCF-binding sites and MIR elements) to build our library. The CTCF-binding sites were already ranked by their predicted binding affinity (Liu et al, 2015, PMID: 25580597). Since the goal of screening was to identify potent barrier elements and CTCF was known as an important player in chromatin loop formation, we focused on sequences with high CTCF-binding affinity and included the top 450 elements in our library. We also included the bottom 50 sequences with low CTCF-binding affinity for comparison. Although MIR elements were not functionally ranked, there were in total only 1,178 such sites predicted across the human genome (Wang et al., 2015, PMID: 26216945), and we included 450 of them into our library, which accounted for more than a third of the entire database. In addition, the technical challenges to long DNA oligo synthesis (e.g, 300 bases) as well as its associated high cost were other practical concerns to limit our library to 1,000 elements. We acknowledged this limitation of our work in the Discussion section, and we believe our work as a proof-of-principle will inspire further studies in this field.

"We note that our library elements were relatively short (i.e., 250 bp) due to the current synthesis limit for oligo pools. Hence, only MIRs and CTCF-binding elements were included for screening mostly due to their compact size, which inevitably limited the scope of current work. Nonetheless, considering various elements (e.g., cHS4, S/MAR, UCOE) remained functional at H1, as well as the large-cargo capability of the integrase (e.g., reporter plasmid carrying S/MAR 1-68 is >10 kb in size), future work would benefit from advanced DNA synthesis technologies to extend the length of synthetic DNA, thus further expanding the library scope."

To evaluate the potential contribution of transcription factor VEZF1 to the barrier activity, we performed a statistical analysis on the occurrence of one putative binding motif (5'-GGGG'-3') in DNA elements significantly enriched in four populations: EGFP-negative, -low, -medium and -high (Figure 5A). When comparing to the top 50 hits in the EGFP-negative population, we found higher occurrence of the target motif in elements enriched in the EGFP-low/medium/high population, with p values smaller than or close to 0.05 (Figure 5C). When the three populations were combined to create the “EGFP-positive” group, the difference in motif occurrence became more statistically significant ($p = 0.024$) between the EGFP-negative ($n = 50$) and EGFP-positive ($n = 179$) population (Figure 5C). This result is consistent with the role of transcription factor VEZF1 in DNA demethylation as previously reported by Dickson et al. (PMID: 20062523). Although in our work we did not provide data directly showing the binding of VEZF1 to library elements during the silencing period, we believe our large dataset ($n \geq 50$) in the revised manuscript provides strong statistical confidence to support our claim, which is in line with the previous model based on the prototypical insulator *chs4*.

In summary, the aim of this work to clarify how barrier insulators may be identified and classified is laudable and points out a missing area of current research. Clearly identifying and designing better insulators for experimental and therapeutic purposes would be of considerable value. Unfortunately, the approach presented includes only preliminary and inconclusive results.

Response: We deeply appreciate the reviewer’s interest and efforts in reviewing our work. The reviewer’s comments and suggestions are highly constructive, and we have taken necessary approaches to the best of our capabilities to address these concerns. We believe the quality of our work has been substantially improved and thus would merit a second round of review.

Reviewer #2

Summary

In this manuscript, the authors introduce a new platform, called SHIELD (Site-specific heterochromatin insertion of elements at lamina-associated domains) for the high-throughput screening of barrier-type DNA elements in human cells by exploiting the naturally occurring heterochromatin spreading inside lamina-associated domains (LADs). The authors hypothesized that selecting LAD locus sites with a strong repressive epigenetic landscape could rapidly silence a reporter gene, which when shielded by potent barrier elements, would retain active expression. To study this they chose sites susceptible to epigenetic silencing to conduct their experiments. This manuscript addresses an important challenge in the field of transgene silencing by identifying barrier DNA elements that can be used to mitigate this challenge.

Response: We appreciate the reviewer’s time and efforts in reviewing our work, and we thank the reviewer for this positive comment on our manuscript.

Comments for authors:

1. On page 4, line 83 the authors state that using PhiC31 integrase circumvents the need for tedious clonal selection. I think the authors need to be care with this statement because utilizing the PhiC31 integrase system doesn’t prevent random integration, it only increases efficiency of integration into the docking site. Scientists will still need to confirm the correct integration at the docking site location and that it was not randomly integrated into the genome.

Response: We agree with the reviewer’s comment and thank the reviewer for pointing out this issue. We believe by “random integration” the reviewer was referring to the integration of donor plasmid at endogenous pseudo *att* sites by the integrase. Indeed, it is possible that even in the presence of the *bona*

vide landing pad artificially inserted at the target site (i.e., E1, H1, H2, H3), the reporter plasmid might still be inserted by the integrase at endogenous pseudo *att* sites that are naturally present in the human genome.

To address this concern, we designed primers (Supplementary Table 5) targeting the top 3 pseudo sites previously identified in the human genome (Chalberg et al., 2006, PMID: 16414067). A similar PCR-based approach was also adopted to evaluate the off-target integration by a different integrase in mammalian cells (Anzalone et al., 2022, PMID: 34887556). Notably, from 10 clones analyzed with confirmed on-target integration at inserted landing pads, we observed no PCR amplicons corresponding to the off-target donor integration at any of the 3 pseudo sites (Supplementary Figure 5), indicating a strong preference of the integrase towards the *bona fide* attP site in the artificial landing pad over endogenous pseudo sites. Hence, we concluded “*Because of the high specificity of SHIELD with no detectable off-target integration (Supplementary Figure 5B), we applied the PuroR polyclonal population directly for analysis without clonal isolation.*”

2. The observation that a second EGFP⁺ population of lower intensity gradually emerges in 14 days is interesting. Have the authors considered taking the study out to 21 days or even longer to see what this population does?

Response: We thank the reviewer for this comment. In the previous manuscript, we only included EGFP expression data up to 14~21 days because gene silencing occurred fast at H1 with the most significant silencing observed in the first 7 days (Figure 2D-G) in the negative control group (N.C.). Interestingly, when the reporter was flanked by protective elements such as the full-length cHS4, we observed a second EGFP⁺ population of lower intensity gradually emerged (Figure 3B), which resulted in the decrease of population MFI over time (Figure 3D) despite EGFP⁺ population percentage (%) remained relatively stable over 14 days (Figure 3C).

As requested by the reviewer, we extended the study to 40 days with a focus on the second EGFP⁺ population of lower intensity. As shown below, in the N.C. group we observed ~12% decrease in EGFP⁺ population percentage (%) from Day 14 to Day 40 with no obvious change in the distribution of the EGFP⁺ population. However, in the full-length cHS4 group we found the second EGFP⁺ population of lower intensity became more and more dominant over time, with almost equal height of peaks detected on Day 40 between the EGFP-low and EGFP-high populations in the histogram. Notably, during this period the EGFP⁺ population percentage (%) remained relatively stable (>90%). These results further indicated that the chromatin insulator cHS4 was highly potent at protecting the reporter gene from being switched off. However, when tested at a highly repressive LAD locus like H1, cHS4 appeared limited at maintaining the expression level of the EGFP⁺ population. We proposed that mRNA degradation at H1 could likely explain the decrease in MFI of the EGFP⁺ population (Supplementary Figure 7). These results are included in our revised manuscript as Supplementary Figure 8B.

Representative EGFP Histograms of Two Populations on Day 14, 28 and 40.

3. The assertion that VEZF1 is recruited to the artificial sequences is not explained and there is no data supporting this claim (page 13 line 290).

Response: We thank the reviewer for this comment. In our newly designed high-throughput screening library we removed all artificial sequences for two reasons: (1) the repetitive sequences in artificial elements could be problematic for oligo synthesis, amplification and next-generation sequencing analysis; (2) artificial sequences with synthetic binding motifs may not recruit transcription factors as intended since the binding sites were putative and additional sequences other than the dG quadruplex may be needed to assist VEZF1 binding. In this regard, we agree with the reviewer that it is not accurate to assume VEZF1 was recruited to the artificial sequences.

We reasoned that the frequency of putative binding motifs in endogenous DNA sequences would be a better proxy for the involvement of corresponding transcription factors than in artificial/synthetic sequences. Hence, in our new library we included 950 endogenous sequences from the human genome with no modifications (Figure 4D). Since the reviewer's question about the contribution of transcription factor VEZF1 to barrier activity is similar to Reviewer #1's last comment, we put our previous answer here for the reviewer's convenience: To evaluate the potential contribution of transcription factor VEZF1 to the barrier activity, we performed a statistical analysis on the occurrence of one putative binding motif (5'-GGGG'-3') in DNA elements significantly enriched in four populations: EGFP-negative, -low, -medium and -high (Figure 5A). When comparing to the top 50 hits in the EGFP-negative population, we found higher occurrence of the target motif in elements enriched in the EGFP-low/medium/high population, with *p* values less than or close to 0.05 (Figure 5C). When the three populations were combined to create the EGFP-positive group, the difference in motif occurrence became more statistically significant (*p* = 0.024) between the EGFP-negative (*n* = 50) and EGFP-positive (*n* = 179) population (Figure 5C). This result is consistent with the role of transcription factor VEZF1 in DNA demethylation as previously reported by Dickson et al. (PMID: 20062523). Although in our work we did not provide data directly showing the binding of VEZF1 to library elements during the silencing period, we believe our large dataset (*n* ≥ 50) in the revised manuscript provides strong statistical confidence to support our claim, which is in line with the previous model based on the prototypical insulator cHS4.

4. The authors have identified low, medium, and high expression levels (Supp Fig. 9), have any studies been done to further separate and analyze each population for epigenetic markers/molecules that may be impacting expression levels?

Response: We thank the reviewer for this comment. Following the removal of selection pressure, we sorted cells into four populations depending on the level of EGFP expression: negative, low, medium and high (Supplementary Figure 12A). Based on the NGS data of populations sorted on Day 3 (see image below, also in Supplementary Figure 13A), we already noticed a distinct distribution of elements in different EGFP-expressing populations. For instance, we observed certain CTCF-family elements (e.g., ID = 1~350, left) were on average more enriched in the EGFP-low population than EGFP-high population. In comparison, certain MIR elements (e.g., ID = 500~750, right) were more enriched in the EGFP-high population than EGFP-low population. This population-dependent distribution of library elements suggested efficient separation of elements according to their barrier activities through SHIELD and FACS. Thus we did not further separate the sorted populations.

Abundance of certain library elements in the plasmid DNA library (pDNA) and sorted populations (day 3)

We focused on the EGFP-high population (Day 15) because elements enriched in this population were more likely to exhibit strong barrier activities. This population was the top ~7% cells from the unsorted population in terms of EGFP expression (Supplementary Figure 12A), and we identified 51 elements significantly enriched in this population from the 1000 library candidates (Figure 5C, High). Although it is possible to narrow down the range of top hits by further separating the EGFP-high population (e.g., sort out the top 1%), we considered it not necessary because it would likely yield a subset of the 51 identified elements. Considering these 51 elements could already be ranked by their fold of change (F.C.) and corresponding p value as shown in the volcano plot (Figure 5C), we anticipated the benefits of further separations would be limited.

For the reviewer's second question about epigenetic markers/molecules that may be impacting expression levels, we assume the reviewer was asking about the change in epigenetic marks caused by active barrier elements. This question is similar to the last comment of Reviewer #1's fourth point, and we would like to respond by reiterating that the focus of our work was to develop and characterize the SHIELD platform and demonstrate its potential for high-throughput screening. We believe in the novelty and significance of our SHIELD system as it represents, to the best of our knowledge, the first platform for high-throughput screening of barrier DNA elements in human cells. Although we did not examine the changes in epigenetic marks caused by different barrier elements in our work, such studies have been performed before in a different context. For instance, Majocchi et al. (PMID: 24071586) studied the mode of action of different epigenetic regulators including CHS4, S/MARs and UCOEs for their ability to prevent transgene silencing at artificial telomeres. We referred to this study in our revised manuscript as it provides relevant information considering both LADs and telomeres are highly repressive domains in human genome.

5. Figures 2E, Supp. Figs. 9, and 10 need a label on the x-axis

Response: We thank the reviewer for pointing out this mistake. We reorganized our figures in the revised manuscript and confirm all axes are correctly labeled in the current version.

Reviewer #3

In this report Zhang et al. describe SHIELD, an assay for screening functional barrier DNA elements in the human genome. The significance of the question being addressed is clearly defined and the authors describe an interesting screening strategy for profiling the activity of putative barrier DNA elements. Unfortunately, there are substantial technical deficiencies that preclude publication of the manuscript in its current form.

Response: We thank the reviewer for his/her efforts in reviewing our work. We particularly appreciate the reviewer's comment on the significance of our work as well as the reviewer's interest in the SHIELD platform. The reviewer's comments below are highly constructive and motivated us to demonstrate the high-throughput potential of SHIELD in a more proper and controlled manner. To address the reviewer's concerns, we reperfomed the screening of a newly designed library (n = 1000) of which the quality and composition were properly evaluated by NGS. For data analysis, instead of focusing on the absolute read abundance (%) of each element in each population, we calculated their relative fold change (F.C.) and corresponding *p* value by comparing against their abundance in the original pDNA library as suggested by the reviewer. These two parameters (F.C. and *p* value) were then used to create the volcano plots shown in Figure 5A, which are commonly adopted to interpret data from large-scale and/or high-throughput screening experiments (e.g., Bao et al., 2018, PMID: 29734295). Detailed responses regarding our efforts to address the technical deficiencies mentioned by the reviewer can be found in our responses below. We believe the quality of our work has been substantially improved and thus would merit a second review.

With the authors having placed the emphasis of the manuscript on the development of a screening platform, the most significant deficiencies fall within construction of the barrier element library. First, during cloning of the library the authors estimate ~10-fold coverage of their plasmid library based on colony-forming units. When building these types of libraries, it is often suggested to aim for ~100-fold coverage based on colony-forming units which raises the concern that not all the elements actually exist in the pDNA library. The authors also fail to provide data describing the actual composition of the pDNA library. They need to sequence the pDNA and evaluate the representation of each element in the library. Importantly, in the absence of this data it is impossible to evaluate enrichment/depletion of library elements in an experiment.

Response: We thank the reviewer for this comment. We constructed the newly designed reporter plasmid (pDNA) library with following improvements. First, we performed two Golden Gate assembly reactions in parallel and obtained 472-fold and 842-fold coverage of the new library, respectively, based on the estimation of colony-forming units. We found that the reason why we obtained a low coverage fold in our previous library cloning was that we used competent cells with low transformation efficiency (i.e., heat-shock transformation). We switched to electrocompetent cells for library construction this time, which helped substantially increase the cloning efficiency and thus the library fold coverage. We continued with the transformants with 842-fold coverage to extract library pDNA by Maxiprep. As an additional step to check the quality of library plasmids, we randomly picked 10 colonies from the *E. coli* transformants and examined the cloning efficiency by colony PCR, which revealed a 90% (9/10) correct insertion rate (Supplementary Figure 11A).

Next, we isolated library pDNA by Maxiprep and determined the composition of pDNA library by NGS. NGS analysis of the pooled plasmids revealed a 98.5% coverage of selected elements (985/1000) in the constructed pDNA library, with a relatively equal distribution of elements according to the cumulative fraction distribution curve (see figure below, also in Supplementary Figure 11B). The positive cloning rate (90%) determined by colony PCR was slightly lower than the library coverage (98.5%) determined by NGS, which should be more accurate considering the low resolution and sensitivity of colony PCR compared to NGS (i.e., only 10 colonies were screened by PCR). Hence, we concluded that our new pDNA library was of high quality and proceeded to the transfection with this pDNA library (Figure 4H). We also used this new information (i.e., composition of the pDNA library) to determine the enrichment/depletion level of library elements in sorted populations, as described in our responses to the reviewer's following comments.

Cumulative fraction distribution curve of elements in pDNA library. AUC: area under the curve. For an ideal library with perfect uniformity ($n = 1000$), $AUC = 500$.

With regards to the screen itself, the data is not properly analyzed and presented. To evaluate the activity of a putative barrier DNA element the authors must compare the abundance of an individual library element in sorted cells to its abundance in the starting pDNA library (data that is not provided). In the absence of this normalization the experiments will always be biased towards elements that just happen to be highly abundant in the library. In fact, the relatively uniform signal from “functional” barrier DNA elements across all time points and sorted populations suggests this to be the case.

Response: We thank the reviewer for pointing out this technical deficiency. Indeed, we noticed that in our current pDNA library, although the vast majority of elements were present with similar abundance, two small groups of elements (e.g., ID=200~250, 450~500) had higher abundance than others in the pDNA library (Supplementary Figure 13A, column 1). These two groups of elements could also be visually detected from the cumulative fraction distribution curve as two “bumps” in this diagonal curve (see figure above, also in Supplementary Figure 11B). Therefore, we agree with the reviewer that it is necessary to normalize the abundance of elements in sorted populations to their abundance in the original pDNA library in order to control for potential biases.

To address this concern and to better control for other potential technical biases, we first generated two “biological replicates” immediately following the removal of selection pressure by splitting the polyclonal cells into two separate populations (Day 0). These two replicates then underwent LAD-induced silencing and cell sorting independently from each other. The results from these two replicates were then used to generate a volcano plot for each population that depicted the enrichment/depletion level of elements with corresponding statistical confidence (i.e., p value). Specifically, in our revised manuscript, instead of focusing on the absolute read abundance (%) of each element in sorted populations, we calculated their relative fold change (F.C.) and corresponding p value by comparing against their original abundance in the pDNA library. These two parameters (F.C. and p value) were used to create the volcano plots (Figure 5A) that are commonly adopted to interpret data from large-scale and high-throughput screening experiments.

The reviewer also mentioned that in our previous screening some elements were highly present in all populations regardless of the EGFP expression level, which was likely caused by technical deficiencies. We believe in our revised work the data quality has been substantially improved after adopting additional approaches as well as a more stringent analysis of NGS results (see Methods). We put here our answer to a similar question above for the reviewer’s convenience: By visually examining the NGS heatmap (Supplementary Figure 13A) we no longer noticed elements that were highly present in all populations regardless of EGFP-expressing level. Instead, for most library elements we found a distribution pattern that

was dependent on EGFP expression level. The image below shows one of many such examples. Despite having a relatively high abundance in the pDNA library, element A was present with high abundance only in the EGFP-medium population sorted on day 3 and was further enriched in this population sorted on Day 15, with low abundance in the EGFP-high population (Day 3 and Day 15). Element B had higher abundance in the EGFP-low population than the EGFP-high population sorted on both days. Element C, by contrast, was mostly enriched in the EGFP-high populations. This population-dependent distribution of library elements suggested efficient separation of elements according to their barrier activities through SHIELD, and it also indicated the technical deficiencies were successfully resolved in our new screening.

Abundance of certain library elements in the plasmid DNA library (pDNA) and sorted populations

Aside from the concerns outlined above, the results from the screen as presented are perplexing. The best performing category of sequences that function as barrier elements appears to be random sequences with 2 of 12 sequences exhibiting strong barrier activity. This raises concerns about SHIELD as an effective screening tool if random sequences give the highest signals. That said, these results are likely artifacts of the library composition as mentioned previously. Based on the control experiments provided in Figures 3B and 3C it seems that the SHIELD approach has the potential to be very insightful if the experiments were to be executed properly.

Response: We thank the reviewer for this comment. In our new library we included 50 randomly generated DNA sequences for control purpose. Following the screening of the new library, we compared the number of random sequences that were significantly enriched in each population sorted on Day 15. As shown below (also in Figure 5A), we found 20 random sequences (black dots) were classified as significantly enriched in the EGFP-negative population. In contrast, only 4, 4 and 6 random sequences were among those elements that were significantly enriched the EGFP-low, -medium and -high population, respectively. In fact, when examining the enrichment level (i.e., fold change and corresponding p value) of these random sequences in these EGFP-positive populations (low/med/high), we found the majority of them were close to the threshold (i.e., located close to the dashed line in the volcano plots) except three sequences enriched in the EGFP-medium population. Hence, random DNA sequences were significantly more enriched towards the EGFP-negative population, supporting that our revised SHIELD screening pipeline could filter out non-functional elements in a high-throughput manner.

Volcano plots showing NGS results of four populations sorted on day 15. Horizontal dashed line: P-value = 0.05. Vertical dash lines: F.C. = 1.5 (right) or 0.67 (left).

We selected 9 elements significantly enriched in the EGFP-high population sorted on Day 15 (indicated in the corresponding volcano plot shown above) for experimental validation. We found eight out of nine (~89%) selected hits exhibited barrier activity that was comparable to the *cHS4_core* in terms of shielding *EGFP* from complete silencing (Figure 5D). Furthermore, two elements, Seq268 (renamed CTCF268) and Seq801 (renamed MIR801), outperformed *cHS4_core* by further elevating *EGFP* expression at H1 as determined by the population MFI (Figure 5E,F). Interestingly, we also noticed that these two elements were the top two hits with the highest fold change (F.C.) determined by NGS from high-throughput screening (volcano plot shown above). More information about these two elements can be found in our revised manuscript (Figure 5G, Supplementary Table 12). In summary, we have taken additional approaches to address the reviewer's concerns. As a result, we strongly believe the quality of our work has been substantially improved and thus would merit a second review.

Reviewers' Comments:

Reviewer #1:
Remarks to the Author:

Reviewer #1

Following on from the original work of Felsenfeld, it is known that the genome is punctuated with elements that can act either to insulate genes from the surrounding influence of positive and negative environmental effects (barrier insulators) or interrupt enhancer-promoter interactions (enhancer blockers). In experimental systems, both types of element protect transgenes from position effects resulting from the chromosomal environment in which a transgene integrates. While the role of CTCF bound elements acting as enhancer blockers has been intensively investigated, their role as barrier insulators has not been studied to the same extent. In this paper, the authors explore this question: How do different DNA elements protect a transgene from silencing when it is introduced into a repressive heterochromatic environment. This is an important outstanding question and if answered would contribute to our understanding of how certain elements can form a barrier to the spread of heterochromatin.

Response: We thank the reviewer for summarizing the background and reiterating the significance of our work. This comment is insightful and echoes with our motivation. We also added some of Felsenfeld's previous studies as references to provide more background of our work.

To address this Zhang et al developed an assay, SHIELD, to functionally screen insulators that block the spreading of heterochromatin. To that end, they create an integrase-based landing pad in pre-selected lamina associated domains (H1, H2 and H3) LADs in the human HCT116 cell line. At each site they integrate a reporter gene tagged with EGFP, and flanked by test barrier elements. EGFP is only detected if the reporter is effectively shielded by the barrier elements from the heterochromatic environment. Sequences of effective barrier elements are then revealed by NGS of DNA from sorted EGFP expressing cells. Using this approach, they identified two novel high affinity binding CTCF sequences with substantial barrier activity and describe a potential role for VEZF1 transcription factor in preventing silencing.

Response: We appreciate the reviewer's time and efforts in reviewing our work. We agree with the major points raised by the reviewer and have significantly revised our manuscript to address these issues. Detailed responses are summarized below.

Major points

1) Although the three test beds (H1, H2 and H3) all classify as LADs they are clearly different forms of heterochromatin. Otherwise, these landing areas are poorly characterised. The only repressive mark assayed is H3K9me3 while there are many others that should have been evaluated, not least H3K27me3, HP1, nuclease/transposase accessibility, DNA Methylation etc. In addition, they have not tested these areas for intrinsic CTCF binding sites which could also influence the results. In summary, the test sites are poorly characterised and it would help their interpretation of the proposed assay to do so.

Response: We thank the reviewer for pointing out the lack of characterization of three heterochromatin test beds. We agree with this comment and have included more epigenetic information to address this issue. Specifically, we included the H3K27me3 mark, DNaseI HS (hypersensitive site), DNA methylation-RRBS (reduced representation bisulfite sequencing) and CTCF ChIP-Seq in the HCT116 cell line around four target sites (Figure 1B). A detailed evaluation of these additional epigenetic marks indeed revealed H1, H2 and H3 as heterochromatin of distinct features, as described below:

1) H3K27me3: a repressive histone modification that often correlates with facultative heterochromatin. We noticed significantly enriched H3K27me3 marks near heterochromatin site H3, but not at H1 or H2. This observation suggests that H3 is likely part of facultative heterochromatin as opposed to the constitutive heterochromatin where H1 is located (e.g., enrichment of H3K9me3). In fact, we found that H3 is located upstream of the human β -globin gene (*HBB*), a gene that is inactive in several cell types but actively transcribed in HCT116 as evidenced by the RNA-Seq track. This observation is also in line with the dynamic nature of facultative chromatin where genes are either silenced or actively transcribed depending on the cell type.

2) DNaseI HS: regions of chromatin that are sensitive to cleavage by the DNase I enzyme. Overall, within the ~400 kb window analyzed, we found more DNaseI HSs at H2 and H3 than H1, indicating a less condensed chromatin structure at H2 and H3 than at H1. This is consistent with the fact that H3 is located upstream of an actively transcribed gene *HBB* in HCT116, which would require a relatively open chromatin structure for gene expression.

3) DNA methylation-RRBS: endogenous DNA methylation. We found abundant CpG islands around E1 but much fewer at three heterochromatin sites H1, H2 and H3. This difference in CpG density is likely due to the difference in gene density between euchromatin and heterochromatin, as CpG islands are typically associated with promoters of genes. Specifically, the majority of CpG dinucleotides near E1 are not methylated (green), indicating an active transcription environment. Although the remaining CpG dinucleotides at E1 appear moderately (orange) or highly methylated (red), most of them belong to the exons of expressed genes, consistent with a previous study suggesting a positive correlation between methylation and exon expression level. However, CpG methylation status at three heterochromatin sites is less informative. Methyl-RRBS signals at H1 indicate low levels of CpG methylation, but given H1 is located in a gene desert, the biological implication of this observation is unclear. For H3, we noticed moderate CpG methylation near genes that are silent (i.e., genes upstream of H3), in line with the transcription repression inside LAD.

4) CTCF ChIP-Seq: intrinsic binding sequences for the transcription factor CTCF. We agree with the reviewer's comment that intrinsic CTCF binding sites at/near the test beds might interfere with our screening, and we thank the reviewer for pointing out this issue. We examined H1, H2 and H3 for intrinsic CTCF binding sites by analyzing the CTCF ChIP-Seq data in HCT116 cells. Fortunately, we found no endogenous CTCF binding sequences within 30 kb distance to target sites H1-H3, suggesting that the CTCF-related barrier activity, if detected, would be contributed by DNA elements from the reporter plasmid integrated via SHIELD.

We did not include heterochromatin protein 1 (HP1) ChIP-Seq as the reviewer suggested because we were not able to find a published database for this information in HCT116 cells, and performing an independent HP1 ChIP-Seq is considered beyond the scope of our work. Nonetheless, considering HP1 typically associates with H3K9me3 marks, the H3K9me3 track could serve as a good indicator of HP1 binding sites near four target sites.

In summary, we included additional epigenetic information as described above, which reveals that H3 is likely part of facultative heterochromatin whereas H1 exhibits typical constitutive heterochromatin features. In comparison, H2 appears in-between on the heterochromatin "spectrum". Based on these additional analysis, we added an extra figure (see below, as shown in Figure 1C) showing the relative chromatin compactness, transcription activity, and putative epigenetic repression level of four selected sites for a direct comparison.

Schematic comparing the levels of epigenetic repression and transcription activity at four sites.

Importantly, the authors have now characterized the regions into which the elements have been integrated by gathering published epigenetic and transcriptional profiles of these regions. I have a significant concern in that they say that HCT116 cells which are derived from a colorectal cancer express the beta globin genes. These genes *in vivo* are expressed only in erythroid cells and no others. Are they sure that the cell line is really HCT 116 and not contaminated by another cell line? If HCT 116 cells do express beta globin this raises a concern that they may represent a very abnormal dysregulated cell line that may not be suitable for this assay. It would be important to clarify this and also if these really are HCT 116 cells expressing beta globin, it would be important to test their positive barriers in another less abnormal cell line or primary cells such as fibroblasts.

In figure 1 do the vertical lines indicate the sites of integration? If so, this should be pointed out in the legend.

2) The second issue concerns the nature of the sequences they initially test. Although CTCF sites are well established insulators, matrix attachment regions (MARs) and Ubiquitous Chromatin Opening Elements (UCOEs) are relatively poorly defined elements and at least to my understanding are not generally accepted as defined classes of insulators. Despite this, all such elements seem to have some activity in their assay. The major problem with the experiments as they stand is that these elements range in size from 1215-3613bp, so it is completely unclear as to which sequences are acting as insulators. The authors do not provide any information on this and do not even mention whether the MARs and UCOEs contain CTCF binding sites.

Response: We thank the reviewer for this comment. We agree that S/MARs and UCOEs are relatively poorly defined and less studied compared to CTCF-based insulators such as *cHS4*. However, certain S/MARs located near the boundaries of active chromatin domains exhibit barrier activity and are classified as barrier-type insulators (Felsenfeld et al., 2002, PMID: 11825869). Another study also included S/MARs as genomic insulators alongside *cHS4* (Goetze et al., 2005, PMID: 15743822). In addition, it was shown the prototypical insulator *cHS4* is also associated with the nuclear matrix (Yusufzai and Felsenfeld, 2004, PMID: 15169959), suggesting a potential overlap in mechanism between *cHS4* and S/MARs.

The S/MAR selected in our work, S/MAR 1-68, could counteract the variegation of transgene expression potentially by adjusting DNA curvature and nucleosome positioning, as well as recruiting transcription factors like SatB1, NMP4 and Hox-like family proteins (Girod et al., 2007, PMID: 17676049). However, CTCF was not listed as a core transcription factor with binding sites in S/MAR 1-68 (Girod et al., 2007), and we did not notice significant CTCF-binding signal when examining the CTCF ChIP-Seq data in HCT116 cells (image below). Hence, S/MAR 1-68 likely functions in a CTCF-independent manner.

RNA-seq and CTCF ChIP-Seq at S/MAR 1-68 element (orange) in HCT116 cells

We included the A2UCOE in our work because it has recently emerged as a potent epigenetic regulator that prevents the variegation of transgene expression and silencing in mammalian cells (Pfaff et al., 2013, PMID: 23307570; Ackermann et al., 2014, PMID: 24290698; Müller-Kuller et al., 2015, PMID: 25605798). The A2UCOE is a methylation-free CpG island from the promoter region of human HNRPA2B1-CBX3 housekeeping genes and can counteract transgene silencing through promoter demethylation, recruiting active histone marks and preventing the propagation of repressive histone marks. There are two CTCF binding sites experimentally determined in A2UCOE (Müller-Kuller et al., 2015, PMID: 25605798), which

were also observed in the CTCF ChIP-Seq data in HCT116 cells (image below). Hence, CTCF likely contributes to the barrier activity of A2UCOE.

This therefore identifies a **region** of the genome that counteracts epigenetic silencing rather than a “barrier element” as such. This would not necessarily identify a distinct family of elements.

RNA-seq and CTCF ChIP-Seq at A2UCOE (blue) in HCT116 cells

In summary, we chose these three elements from distinct families with the goal to compare their activities under the same highly repressive chromosome context (i.e., inside the LAD) side by side. Such a comparison has not been performed before due to technical infeasibility, but could be achieved with our SHIELD platform. We note that currently there is a lack of terminology that properly defines or classifies these elements. As a result, they are sometimes referred to as chromatin modifying elements, epigenetic regulators, or simply “anti-silencing” elements. Nonetheless, the lack of proper terminology does not affect the quality of our experiment results, which clearly suggest the full-length cHS4 as the best performing element in preventing epigenetic silencing at a highly repressive LAD locus (Figure 3C-F). We also included the following sentences in the main text for better clarification:

“The cHS4 element has been well studied as a chromatin insulator with potent barrier activity mediated by transcription factors including CTCF, VEZF1 and USF. Certain SMAR elements are also classified as insulators and can increase transgene expression at repressive chromatin by binding to transcription factors such as special (A+T)-rich binding protein 1 (SATB1), nuclear matrix protein 4 (NMP4) or CTCF. UCOEs are commonly referred as “anti-silencing elements” instead of “chromatin insulators”, and they can protect transgene(s) from epigenetic silencing and variegation in mammalian cells. The selected A2UCOE contains two CTCF binding sites, which likely contribute to its reported anti-silencing activity. However, the barrier/anti-silencing activities of these elements have not been compared side by side at the same chromosome context, especially inside highly repressive LADs.”

3) The third point concerns analysis of the experimental data. Figure 3F illustrates the most relevant findings, but how were these data acquired, analysed and normalised? Why are some of the sequences very highly present in all of the populations/groups (68, 55), whereas others are completely absent (22,23)? This could most easily be explained by a technical bias rather than a biological phenomenon, and means these results are impossible to interpret. It cannot be explained by cells simply being negative for GFP (these cells are not sorted and sequenced), because Supplementary Figure 9A shows that this population is very small at day 3, and yet there are many sequences absent from day 3. This could be explained either by integration efficiency or biases introduced during the PCR amplification step. It is crucial to control for these biases.

Response: We thank the reviewer’s comment on the technical details of our screening process and data analysis. To fully address this concern as well as the reviewer’s final comment on the small size of our previous library (n~100), we completely re-designed our library and expanded it to 1000 DNA elements (Figure 4D), which represents a 10-fold increases in the library size. A detailed description and comparison of elements in the new library can be found in our revised manuscript (Figure 4D-F). Briefly, the new library consists of 500 CTCF-binding sequences (including 450 high-affinity and 50 low-affinity binding sites), 450 MIR elements (including 30 MIRs located within 5 kb to LAD boundaries and 420 randomly picked MIRs) and 50 randomly generated DNA sequences for control purpose.

We reconstructed the reporter plasmids library (pDNA) and repeated high-throughput screening with the revised workflow shown in Figure 4H. We adopted three additional approaches to better control for potential technical biases that may affect the screening outcome. First, we determined the composition of

the pDNA library by NGS, which revealed a relatively equal distribution of elements (Supplementary Figure 11B and 13A) with a coverage of 98.5% of the selected 1000 elements. Secondly, immediately following the removal of selection pressure, we split the polyclonal cells into two separate populations (day 0), which served as two “biological replicates” that underwent LAD-induced silencing and cell sorting independently from each other. The results from these two biological replicates were then used to generate the volcano plots shown in Figure 5A. Thirdly, instead of simply focusing on the absolute read abundance (%) of each element in sorted populations, we calculated their relative fold change (F.C.) and corresponding p value by comparing against their abundance in the original pDNA library. Each element was assigned with two parameters (F.C. and p value) which were used to create the volcano plots shown in Figure 5A, a commonly adopted method to interpret data from large-scale and/or high-throughput screening experiments (e.g., Bao et al., 2018, PMID: 29734295).

By adopting these additional approaches as well as a more stringent analysis of NGS results (see Methods section for NGS sample preparation, data processing and analysis), we believe our data quality has been significantly improved. For instance, by visually examining the NGS heatmap (Supplementary Figure 13A) we no longer noticed elements that were highly present in all populations regardless of EGFP-expressing level. Instead, for most library elements we found a distribution pattern that was dependent on EGFP expression level. The image below shows one of many such examples. Despite having a relatively high abundance in the pDNA library, element A was present with high abundance only in the EGFP-medium population sorted on day 3 and was further enriched in this population sorted on day 15, with low abundance in EGFP-high populations (day 3 and day 15). Element B had higher abundance in EGFP-low populations than the EGFP-high populations sorted on both days. Element C, by contrast, was mostly enriched in EGFP-high populations.

Abundance of certain library elements in the plasmid DNA library (pDNA) and sorted populations

Furthermore, we found that the vast majority of elements were present in at least one population sorted on day 3 (Supplementary Figure 13A) except the ones that were not present in the initial pDNA library (coverage = 98.5%). In summary, we believe we have taken necessary and effective approaches to address the reviewer’s concerns on the technical fitness of our high-throughput screening assay. We identified two novel elements with barrier activities better than the `chs4_core` element following the screening of our newly designed library (Figure 5D-G). More detailed information can be found in our revised manuscript.

I agree that this is an improved screen but it is quite limited in the new biological information provided.

4) Point 4 concerns some of the mechanistic conclusions of their findings. The authors claim they can distinguish between transcription and translation dependent silencing in their assay. Although it is true that both these processes affect the cellular EGFP levels, it is not clear how the authors can distinguish between the two. Just the level and spread of EGFP levels can be due to many factors, such as the half-life of the protein which means some fluorescence can still be detected if the silencing of the locus has just started. The authors note that rixosome-triggered degradation might play a role, but this is pure speculation. It would be interesting to mention in the discussion as a possibility, but does not belong in the main body of the text

or Figure 2D. If they want to claim any rixosome involvement they have to perform a ChIP for rixosome subunits (TEX10 or MDN1) and show that it's present at their integration site, preferably in higher levels after more days of puromycin withdrawal than in the beginning. They could also show that nascent transcription is still present but protein levels are down, if they want to claim regulation on a translational/RNA stability level. The authors have made no attempt to evaluate changes in the chromatin or DNA methylation in silenced or non-silenced integrants to determine how these elements are being silenced or protected from silencing.

Response: We thank the reviewer for this detailed comment, and we appreciate his/her suggestions on the experimental design. To explain the silencing pattern that was distinct from the all-or-none silencing phenomenon as we observed in our work, we proposed a model that took into account gene silencing as the translation level. Based on this model, transcriptional silencing is reflected by the percentage of EGFP-negative population (i.e., complete EGFP silencing), whereas translational silencing is indicated by the decrease in the MFI of the EGFP-positive population. Hence, we suggested in our original submission that SHIELD could distinguish transgene silencing at these two levels. We agree with the reviewer that the half-life of EGFP might influence the fluorescence intensity detected at a given time point. However, throughout our study, we intentionally set the time interval between two flow cytometry analyses to be at least 3 days (72 hrs) apart, which is significantly longer than the estimated half-life of GFP (~24 hrs). Therefore, the difference in EGFP intensity detected at two time points (e.g., Day 1 vs. Day 4, or Day 1 vs. Day 7) should be predominantly caused by epigenetic silencing.

We agree that the proposed model was partially based on a recent study (Zhou et al., 2022, PMID: 35355014), and our work did not directly confirm the involvement of rixosome during epigenetic silencing at H1. Nevertheless, we believe this model provides the most reasonable explanation for the observed silencing pattern. For scientific rigor, we toned down our claim in the revised manuscript and moved the corresponding figure from the main body to the supplementary information (Supplementary Figure 7).

As to the reviewer's last comment on efforts to evaluate changes in chromatin or DNA methylation status, we would like to point out that the focus of our work was to develop and characterize the SHIELD platform and demonstrate its potential for high-throughput screening. To the best of our knowledge, such a platform has not been established before particularly in human cells. Additional studies are warranted to reveal the mechanisms by which active elements protect the reporter from epigenetic silencing at the highly repressive LAD locus H1, but in our opinion they are beyond the scope of our current work. On this front, Majocchi et al. (PMID: 24071586) studied the mode of action of different epigenetic regulators including cHS4, S/MARs and UCOEs for their ability to prevent transgene silencing at artificial telomeres, and revealed epigenetic signatures associated with the reduced silencing including increased acetylation marks and decreased deposition of repressive marks on nearby nucleosomes. We cited this study in our revised manuscript as it provides relevant information considering both LADs and telomeres are highly repressive domains in human genome.

The final screen involves only ~100 sequences which far too small a group from which to draw conclusions. Highlighting the role of VEZF based on previous conclusions from the Felsenfeld lab is speculative and they provide no experimental evidence for a role for VEZF.

Response: We thank the reviewer for this comment and we agree that our previous library was relatively small in size. As we described in our response to the reviewer's third point, we expanded our library by an order of magnitude ($n = 1,000$) and reperformed our screening assay with the new library. Although it is possible to further increase the library size as the integrase-based screening could accommodate large library of variants ($n = 12,000$ demonstrated by Cao et al., 2021, PMID: 34230498), we limited our library to 1000 elements mainly due to the limited choices of candidate elements. Currently, there is limited knowledge of what constitutes a functional barrier element, and nor are there sufficient barrier elements with validated activities that could be adopted to train a computational model. As a result, we relied on two published databases (genome-wide CTCF-binding sites and MIR elements) to build our library. The CTCF-

binding sites were already ranked by their predicted binding affinity (Liu et al, 2015, PMID: 25580597). Since the goal of screening was to identify potent barrier elements and CTCF was known as an important player in chromatin loop formation, we focused on sequences with high CTCF-binding affinity and included the top 450 elements in our library. We also included the bottom 50 sequences with low CTCF-binding affinity for comparison. Although MIR elements were not functionally ranked, there were in total only 1,178 such sites predicted across the human genome (Wang et al., 2015, PMID: 26216945), and we included 450 of them into our library, which accounted for more than a third of the entire database. In addition, the technical challenges to long DNA oligo synthesis (e.g, 300 bases) as well as its associated high cost were other practical concerns to limit our library to 1,000 elements. We acknowledged this limitation of our work in the Discussion section, and we believe our work as a proof-of-principle will inspire further studies in this field.

“We note that our library elements were relatively short (i.e., 250 bp) due to the current synthesis limit for oligo pools. Hence, only MIRs and CTCF-binding elements were included for screening mostly due to their compact size, which inevitably limited the scope of current work. Nonetheless, considering various elements (e.g., cHS4, S/MAR, UCOE) remained functional at H1, as well as the large-cargo capability of the integrase (e.g., reporter plasmid carrying S/MAR 1-68 is >10 kb in size), future work would benefit from advanced DNA synthesis technologies to extend the length of synthetic DNA, thus further expanding the library scope.”

To evaluate the potential contribution of transcription factor VEZF1 to the barrier activity, we performed a statistical analysis on the occurrence of one putative binding motif (5'-GGGG'-3') in DNA elements significantly enriched in four populations: EGFP-negative, -low, -medium and -high (Figure 5A). When comparing to the top 50 hits in the EGFP-negative population, we found higher occurrence of the target motif in elements enriched in the EGFP-low/medium/high population, with p values smaller than or close to 0.05 (Figure 5C). When the three populations were combined to create the “EGFP-positive” group, the difference in motif occurrence became more statistically significant ($p = 0.024$) between the EGFP-negative ($n = 50$) and EGFP-positive ($n = 179$) population (Figure 5C). This result is consistent with the role of transcription factor VEZF1 in DNA demethylation as previously reported by Dickson et al. (PMID: 20062523). Although in our work we did not provide data directly showing the binding of VEZF1 to library elements during the silencing period, we believe our large dataset ($n \geq 50$) in the revised manuscript provides strong statistical confidence to support our claim, which is in line with the previous model based on the prototypical insulator cHS4.

I disagree that they can confidently make this conclusion about the importance of VEZF1.

In summary, the aim of this work to clarify how barrier insulators may be identified and classified is laudable and points out a missing area of current research. Clearly identifying and designing better insulators for experimental and therapeutic purposes would be of considerable value. Unfortunately, the approach presented includes only preliminary and inconclusive results.

Response: We deeply appreciate the reviewer’s interest and efforts in reviewing our work. The reviewer’s comments and suggestions are highly constructive, and we have taken necessary approaches to the best of our capabilities to address these concerns. We believe the quality of our work has been substantially improved and thus would merit a second round of review.

Although the manuscript is improved, I do not think the biological conclusions merit publication in Nature Communications. It may be better to present their method in a journal devoted to such technical reports and await the fuller characterization of new elements identified by this method.

Reviewer #2:

Remarks to the Author:

The authors have addressed my concerns.

Reviewer #3:

Remarks to the Author:

In the revised manuscript the authors have addressed my concerns regarding the analysis of their data. I believe the revised manuscript is greatly improved and warrants publication. My only comment is that NGS data needs to be deposited in GEO/SRA prior to publication.

Reviewer #1

1) Importantly, the authors have now characterized the regions into which the elements have been integrated by gathering published epigenetic and transcriptional profiles of these regions. I have a significant concern in that they say that HCT116 cells which are derived from a colorectal cancer express the beta globin genes. These genes in vivo are expressed only in erythroid cells and no others. Are they sure that the cell line is really HCT 116 and not contaminated by another cell line? If HCT 116 cells do express beta globin this raises a concern that they may represent a very abnormal dysregulated cell line that may not be suitable for this assay. It would be important to clarify this and also if these really are HCT 116 cells expressing beta globin, it would be important to test their positive barriers in another less abnormal cell line or primary cells such as fibroblasts.

Response: We thank the reviewer for his/her efforts in reevaluating our work, particularly for pointing out that normal HCT116 cells should not express the beta globin gene. After carefully reexamining the published RNA-seq data in HCT116 cells, we found that **our previous statement in the main text was incorrect** (i.e. “*H3 is upstream of the human β -globin gene (*HBB*) that is inactive in several cell types but actively transcribed in HCT116*”). As explained below, this mistake was due to our inaccurate interpretation of RNA-seq at the beta-globin gene cluster located near H3, as well as our improper display of the RNA-seq track in the UCSC Genome Browser.

As shown below, the top image (#1) was included in our manuscript (Figure 1B, H3) to reflect the transcription activity around target site H3 in HCT116 (scale bar = 200 kb). Initially, we noticed an elevated level of transcription at the 3' side of H3 that is close to the *HBB* gene, and we jumped to the conclusion that *HBB* is actively transcribed in HCT116 without further confirming its transcription activity by zooming into the beta-globin gene cluster. In fact, a detailed view of RNA-seq track at the beta-globin gene cluster (image #2, scale bar = 20 kb) reveals that the ***HBB* gene is not actively expressed in HCT116** cells, consistent with the erythroid-specific expression profile of *HBB* as the reviewer pointed out.

We also noticed that while *HBB* and *HBD* show no active expression in HCT116, three other beta-globin genes (*HBG1*, *HBG2* and *HBE1*) appear to be expressed to a certain extent according to RNA-seq shown in image #2. This observation prompted us to check the data view option we selected to display the RNA-seq track in Genome Browser. We found that in both image #1 and #2 we selected the “manually set vertical viewing range” option to display RNA-seq track with Min = 1 and Max = 10. In this case, RNA-seq signals greater than 10 are cut off from the y-axis, resulting in loss of information, especially when viewing RNA-seq of multiple genes within the same window.

Therefore, we changed the data viewing option to “auto-scale to data view”, which automatically scales to a range defined by the minimum and maximum data points in the current view. By viewing RNA-seq in this way (image #3), we found that only the *HBE1* gene (epsilon-globin) is significantly highly expressed among the beta-globin gene cluster in HCT116 cells. This is consistent with a previous study suggesting that the active expression of *HBE1* contributed to the radiation resistance of colorectal cancer cell lines including HCT116 (PMID: 30965648). We also included the absolute RNA-seq signal value of five beta globin genes and one housekeeping gene *GAPDH* in HCT116 and K562 cells for reviewer’s reference (Table 1). The K562 cell line is included for comparison as it represents a distinct cell type (i.e., lymphoblast cells) from HCT116, and the high-level expression of gamma-globin (*HBG1* and *HBG2*) in K562 cells has been well documented and studied (e.g., PMID: 30012865). We further examined another RNA-seq database from ENCODE (ENCSR698RPL) and found a similar expression pattern of the beta-globin gene cluster in HCT116, consistent with what is shown in image #3 (ENCSR000CWM).

In summary, we mistakenly stated that *HBB* was actively expressed in HCT116 due to our incorrect interpretation and display of RNA-seq data. We apologize for the confusion it caused. We have corrected this mistake in our revised manuscript and updated RNA-seq tracks in Figure 1B accordingly. This clarification should address the reviewer's concern that HCT116 may represent an abnormal dysregulated cell line.

In fact, we want to point out that HCT116 is commonly used for biological studies and is recommended as a Tier 2 cell line by the 4D Nucleome which funded our work. Numerous studies have been published using HCT116 cells, such as studies on replication timing (PMID: 33888635), dynamics of lamina-associated DNA (PMID: 32893442) and topologically associating domains (PMID: 32841603). We obtained our original HCT116 stock from the 4D Nucleome recommended provider (ATCC) and have since then strictly followed the cell culture protocol. We are confident to claim that we did not encounter any cross contamination from another cell line throughout our study. Additionally, we want to emphasize that our screening was performed predominantly at the H1 test bed, not H3. H1 is located inside a constitutive LAD, and constitutive LADs are shown to be conserved across cell types (PMID: 23124521). Hence, we anticipate that the barrier activity detected at a constitutive LAD region such as H1 should likely be retained in other cell types.

In addition to our response above, we would like to provide the following information to further explain our choice of HCT116 for this study:

- 1) Near-diploid karyotype. SHIELD-based screening necessitates the integration of reporter plasmids with the same copy number to exclude copy number-dependent expression variation. Hence, it is important to obtain chassis cell lines with single-copy landing pad insertion. The near-diploid karyotype of HCT116 makes it easier to distinguish the copy number of inserted landing pad by genotyping PCR: no insertion = only WT band, single-copy insertion = WT band + Knock-in band, two-copy insertion = only Knock-in band (see gel images in Supplementary Figure 2).
- 2) Difficulty in CRISPR knock-in in mouse embryonic stem cells (mESCs). Initially, we also included F123 mESCs in our study alongside HCT116, reasoning that transgene silencing could be more prominent in stem cells due to their tight epigenetic regulation profile. However, we encountered great technical difficulty in creating F123 knock-in chassis clones at heterochromatin sites, which was likely caused by a combination of low transfection efficiency and low CRISPR editing rate at compact lamina-associated domains in stem cells. In fact, even for HCT116 cancer cells for which transfection was fairly efficient, we only obtained one correct clone from 34 analyzed clones (for H1) using our optimized knock-in method.
- 3) Suitable for imaging studies. As part of the 4D Nucleome consortium, we intended to maximize the usage of our cell lines for other projects going on in the group, many of which involve the imaging analysis of cells to study genome spatial organization. HCT116 cells are well suited for microscope experiments due to their relatively large size and flat morphology.

2) In figure 1 do the vertical lines indicate the sites of integration? If so, this should be pointed out in the legend.

Response: Yes, all vertical lines in Figure 1B indicate the location of the inserted landing pad and thus mark the integration sites of SHIELD reporters. We thank the reviewer for pointing out this missing information, and we have added the description in the legend in our revised manuscript.

3) This therefore identifies a *region* of the genome that counteracts epigenetic silencing rather than a “barrier element” as such. This would not necessarily identify a distinct family of elements.

Response: We thank the reviewer for this detailed comment. We believe the reviewer was referring to the ubiquitous chromatin opening element (UCOE) here. We agree with the reviewer that it is not appropriate to include UCOEs as “barrier elements” because whether UCOEs function by establishing chromatin barriers in a fashion similar to *chs4* remains undetermined. To account for this ambiguity in terminology, we changed the title of the corresponding subsection to “Known barrier **and anti-silencing** elements are active inside the LAD”, since UCOEs are more commonly referred to as anti-silencing elements.

As to the distinction between “region” and “element” as the reviewer carefully pointed out, we intend to keep our description of UCOEs consistent with existing literature in order to avoid potential confusion for future readers. In case the reviewer missed it, in our previously revised manuscript we included the following sentences to further clarify UCOEs: “*UCOEs are commonly referred as ‘anti-silencing elements’ instead of ‘chromatin insulators’, and they can protect transgene(s) from epigenetic silencing and variegation in mammalian cells. The selected A2UCOE contains two CTCF binding sites, which likely contribute to its reported anti-silencing activity.*”

Furthermore, as for the reviewer’s comment that “This would not necessarily identify a distinct family of elements”, we would like to respond by clarifying that it was not within our aim to identify new UCOEs in this work. Rather, in our screening we focused on two complementary classes of DNA elements: CTCF-binding sites and MIRs. The A2UCOE was used in our work **only for the comparison study** against the full-length *chs4* and S/MAR8 element, which preceded our screening of candidate DNA elements with unknown barrier activities by SHIELD. The discovery of novel UCOEs has been reported previously (doi.org/10.1101/626713), but it is irrelevant to our work here.

4) I agree that this is an improved screen but it is quite limited in the new biological information provided.

Response: We thank the reviewer for recognizing our efforts to improve our work. To respond to this comment, we would like to provide a brief summary of the biological information we consider new and significant for a more objective evaluation of our work in this regard.

(1) Our study reveals that the serine integrase retained its advantage over the CRISPR/Cas9 system at highly compact heterochromatin such as LADs. In recent years, the large serine integrase has been gaining popularity for targeted DNA integration in mammalian cells and was recently adopted as a key module for the development of Twin Prime (PMID: 34887556) and PASTE (PMID: 36424489). However, the targeting scope of previous works was limited because only easy-to-edit genomic sites were selected for testing, i.e., housekeeping genes at open euchromatin. It is known that chromatin architecture

influences the editing outcome of genome editors like Cas9 (PMID: 26987018, 30201707). In this respect, our work represents the first effort to systematically evaluate the performance of serine integrase at structurally distinct chromatin (i.e., heterochromatin of varying compactness) in the human genome, thus completing the targeting scope of the integrase-based strategy for DNA integration in mammalian cells. We kindly argue that this new information **regarding the robustness of the serine integrase** should not be overlooked.

- (2) Our work provides strong and direct evidence showing that epigenetic silencing in human cells is **not simply an “all-or-none” phenomenon**. Previous studies including seminal works by Hathaway et al. (PMID: 22704655) and Bintu et al. (PMID: 26912859) reported silencing as all-or-none events in mammalian cells. These studies, however, were limited by their adopted method to **artificially induce** the silencing of reporter genes **at transcriptionally active sites** through targeted recruitment of transcription repressor. We believe these approaches, although effective, were inadequate to fully recapitulate gene silencing in mammalian cells as they failed to take into account the potential **context dependency** of gene silencing. For instance, the kinetics of artificially induced gene silencing at an otherwise transcriptionally active region (image below, right) might be different from that of naturally occurring silencing inside highly repressive LADs (image below, left), as the latter may involve a concerted action of multiple silencing machineries (e.g., rixosome for mRNA degradation) that preferentially localize to repressive nuclear compartments such as the lamina.

Our work enables efficient analysis of gene silencing **in situ** (i.e., directly under the context of highly repressive chromosome domains), thus complementing previous studies that focused on artificially induced gene silencing. Hence, our study is informative in this regard as it strongly indicates that gene silencing in mammalian cells employs diverse mechanisms beyond the all-or-none kinetics as previously reported.

- (3) Our work reveals that the rate of epigenetic silencing was positively correlated with the density of repressive histone mark H3K9me3 in human cells. While this kinetic correlation may seem obvious, there is a lack of evidence that directly supports this claim in human cells. In previous studies, epigenetic marks such as H3K9me3 near the reporter integration site were either uncharacterized (e.g., random integration by viral vectors) or could not be fine tuned (e.g., induced silencing by chromatin repressors). By contrast, our work examined the silencing of the same reporter gene at epigenetically distinct endogenous loci

that we intentionally selected with varying degree of H3K9me3 marks. Consistent with a recent study in yeast (PMID: 34035174), our work suggests that **the rate of heterochromatin propagation** in human cells is also likely dependent on the density of H3K9me3.

- (4) Our work calls for more caution to **distinguish the barrier activity from the enhancer-blocking activity** of a given DNA insulator. We tested the barrier activity of three CTCF-binding sequences (A2, A4 and E2) previously identified as highly potent enhancer-blocking elements (PMID: 25580597). These three compact elements (243-302 bp) were shown to be much more potent (> 6-fold) than the full-length cHS4 (1.2 kb) element in the enhancer-blocking assay. However, our SHIELD-based assay revealed that these elements exhibited only weak to no barrier activity compared to the full-length cHS4 (Figure 4B-C), highlighting the discrepancy between these two properties of a given DNA insulator. This information is also critical for synthetic biologists because it clarifies that these elements (A2, A4, E2), although compact, should not be adopted to replace the full-length cHS4 element when the goal is to minimize epigenetic silencing. In fact, due to the lack of such information, a recent study (PMID: 33510302) mistakenly chose the A2 element over cHS4 in its design to minimize epigenetic silencing, as the authors stated: *“while our model system employed chromatin insulators ... this was not sufficient to isolate our transcription unit from being silenced ... we employed the A2 chromatin insulator that has previously been shown to have a greater insulating capacity over the canonical cHS4 chromatin insulator.”*
- (5) Our work also points out the **important role of MIR elements in establishing chromatin barriers**. Previous studies on DNA insulators in mammalian genomes (both enhancer-blocking and barrier-type elements) largely focused on CTCF-binding elements, while MIRs received significantly less attention by comparison due to their lack of CTCF-binding motifs. To the best of our knowledge, the most recent systematic study on the insulating role of MIRs was reported in 2015 by Wang et al. (PMID: 26216945). However, only the enhancer-blocking activity of MIRs was tested in this work while the barrier activity was left unexamined due to the lack of an efficient assay. Our work fills up this gap in knowledge by screening ~450 genomic MIR elements and validated four novel MIRs with barrier activities comparable to or greater than the cHS4_core (Figure 5D-E). Furthermore, this information also indicates that there exist diverse types of barrier elements in the human genome that could function in a CTCF-dependent or -independent manner, as recently reviewed by Lawson et al. (PMID: 37286742).

In summary, we believe our work provides sufficient new biological information along with significant technical novelty that merit its publication in Nature Communications. While preparing our response to the reviewer’s comment, we realized some of the points mentioned above were not adequately discussed in our manuscript. Hence, we added the following paragraphs in the discussion section to reflect our response here:

- 5) I disagree that they can confidently make this conclusion about the importance of VEZF1.

Response: We thank the reviewer for this comment. The role of VEZF1 in chromatin barrier function has been studied before using the cHS4 as a model insulator (PMID: 22308494, 20062523). Following this information, we sought to test if our data could provide additional support to the existing model by analyzing the putative binding motifs present in enriched DNA elements. We agree with the reviewer that this approach was not adequate enough to reach a definite conclusion regarding the actual involvement of VEZF1 in the barrier function. We therefore revised our previous claim and describe VEZF1 as a potentially important instead of key player in establishing the chromatin barrier.

6) Although the manuscript is improved, I do not think the biological conclusions merit publication in *Nature Communications*. It may be better to present their method in a journal devoted to such technical reports and await the fuller characterization of new elements identified by this method.

Response: We thank the reviewer for recognizing our efforts to improve this work. To respond, we want to emphasize that our study essentially consists of three parts: the establishment of SHIELD platform, the analysis of silencing kinetics at epigenetically distinct loci, and the high-throughput screening of DNA elements via SHIELD. We appreciate the reviewer's positive comment on the technical novelty and significance of our study. However, we believe that presenting our work in the format of a technical report would significantly understate the importance and breadth of our study, as it will inevitably overlook the new biological information we intend to convey (as summarized above). Hence, we respectfully argue that, instead of focusing only on its technical contribution or biological conclusions, it would be better to adopt an holistic evaluation of our work by considering it as a whole.

Importantly, we believe our manuscript aligns well with the scope of *Nature Communications* and is of great interest to a broad research community. For instance, our strategy is generally applicable, and SHIELD can also be adapted for studies that aim to mitigate transgene silencing via different approaches or to rewire LADs in human cells. Moreover, the silencing kinetics reported in our study will likely spark further discussions and investigations on epigenetic silencing at different contexts in mammalian genomes. Furthermore, our work can also inspire synthetic biologists to build artificial DNA elements with both compactness and potent barrier activities, potentially with the help of machine learning-guided strategies. As to the reviewer's comment on the fuller characterization of new elements, we have validated the barrier activity of top hits in our previously submitted manuscript and included sequence feature analysis focused on two proposed transcription factors. We consider the functional dissection of new elements beyond the scope of our current work, and we will further characterize the new barrier elements in our follow-up publication.

Reviewer #2

The authors have addressed my concerns.

Response: We thank the reviewer for reevaluating our manuscript. We are glad our revised work addressed the reviewer's concerns.

Reviewer #3

In the revised manuscript the authors have addressed my concerns regarding the analysis of their data. I believe the revised manuscript is greatly improved and warrants publication. My only comment is that NGS data needs to be deposited in GEO/SRA prior to publication.

Response: We highly appreciate the reviewer's efforts in reevaluating our manuscript and we thank the reviewer for his/her positive comment. To address the reviewer's concern, we deposited our NGS data in GEO with the accession number GSE236198. We selected Aug 1, 2023 as the release date but can certainly accommodate an early release should the reviewer request.